

# Long-term satellite trends of European lower-tropospheric ozone from 1996 – 2017

Matilda A. Pimlott[1], Richard J. Pope[1,2], Brian J. Kerridge[3,4], Richard Siddans[3,4], Barry G. Latter[3,4], Wuhu Feng[1,5], Martyn P. Chipperfield[1,2]

[1]School of Earth and Environment, University of Leeds, Leeds, LS2 9JT, UK
[2]National Centre for Earth Observation, University of Leeds, Leeds, LS2 9JT, UK
[3]Remote Sensing Group, STFC Rutherford Appleton Laboratory, Chilton, Oxfordshire, OX11 0QX, UK
[4]National Centre for Earth Observation, STFC Rutherford Appleton Laboratory, Chilton, Oxfordshire, OX11 0QX, UK
[5]National Centre for Atmospheric Science, University of Leeds, Leeds, LS2 9PH, UK

*Correspondence to*: Richard J. Pope (r.j.pope@leeds.ac.uk)

**Abstract.**

Tropospheric ozone ($O_3$) is a harmful secondary atmospheric pollutant and an important greenhouse gas. Satellite records have shown conflicting long-term tropospheric ozone trends over the globe, including Europe. Here, we present an in-depth analysis of lower-tropospheric sub-column $O_3$ (LTCO$_3$, surface – 450 hPa) records from three ultraviolet (UV) sounders produced by the Rutherford Appleton Laboratory (RAL): the Global Ozone Monitoring Experiment (GOME, 1996-2010), Scanning Imaging Absorption Spectrometer for Atmospheric Chartography (SCIAMACHY, 2003-2011) and Ozone Monitoring Instrument (OMI, 2005-2017). Overall, GOME and SCIAMACHY have negative trends of approximately -0.2 DU yr$^{-1}$ across their respective full records, while OMI indicates a negligible trend. The TOMCAT 3-D chemical transport model was used to investigate processes driving simulated trends and try to identify possible reasons for discrepancies between the satellite records. However, the model trends generally showed negligible change in LTCO$_3$, even when spatiotemporally co-located to the satellite level-2 swath data and convolved by averaging kernels. Model sensitivity experiments with the emissions or meteorology fixed to 2008 values aimed to isolate the impact of these processes on the simulated LTCO$_3$ trend. Overall, the experiments highlighted a long-term steady balance in these processes with small positive trends (<0.1 DU yr$^{-1}$) between 1996 and 2008 and then small negative trends (>-0.1 DU yr$^{-1}$) between 2008 and 2017. As a result, it is difficult to detect a robust and consistent linear trend in European lower tropospheric $O_3$ between 1996 and 2017, which is masked by large inter-annual variability in the model, ozonesonde and UV satellite instrument records.

## 1 Introduction

Tropospheric ozone ($O_3$) is both detrimental to air quality and an important short-lived climate forcer (e.g. Monks et al., 2015). At the surface, $O_3$ is harmful to human health as it is a strong oxidant, with an estimated 24,000 premature deaths attributed to acute $O_3$ exposure across Europe in 2020 (European Environment Agency, 2022). It is also damaging to plants and reduces crop yields, which is estimated to have caused global economic damage in the region of US$14 - US$26 billion in 2000 (Van



Dingenen et al., 2009). Tropospheric $O_3$ is a significant greenhouse gas (GHG), with an estimated effective radiative forcing of 0.47 (0.24-0.70) W m$^{-2}$ between 1750 – 2019, dominated by changes in tropospheric $O_3$ (IPCC, 2021; Forster et al., 2021; Skeie et al., 2020). It is a secondary pollutant, produced through reactions involving precursor nitrogen oxides ($NO_x$, referring

to nitrogen dioxide ($NO_2$) and nitric oxide (NO)) and volatile organic compounds (VOCs) in the presence of sunlight. Despite anthropogenic emissions of these precursor gases declining in the last 20 years (European Environment Agency, 2022), in 2021 an estimated 10% of the European urban population was exposed to $O_3$ concentrations above the European Union (EU) standards and 94% above the World Health Organization (WHO) guidelines (European Environment Agency, 2023). These persistent exceedances across Europe highlights the need for further study into how near-surface $O_3$ is changing over time.


Satellite retrievals of tropospheric trace gases present an opportunity to enhance our knowledge of atmospheric composition on larger spatial scales (e.g. global or regional) than other observations. Trends in satellite tropospheric $O_3$ retrieved from different instruments have been shown to not be consistent for some regions around the world. Gaudel et al. (2018) presented long-term trends for several satellite tropospheric column $O_3$ products, finding a range of trends from -0.50 DU yr$^{-1}$ to +0.16

DU yr$^{-1}$ for the 30°N – 60°N latitude band which includes Europe. These inconsistent trends are from instruments, using different measurement technique/spectral ranges, e.g. ultraviolet (UV) and infrared (IR), which have different attributes, e.g. spatial coverage, resolution and vertical sensitivity. They also use different retrieval schemes, which means different state-vectors, apriori information, radiative transfer models, tropopause definitions and cloud filters, and data are presented for different vertical ranges and across different time periods between 1996 – 2016. So, there is a need to study these records in

more depth, especially those from similar instruments (e.g. UV here), from the same retrieval scheme and over the same time period, to minimise the most obvious sources of difference between the records. Aside from Gaudel et al. (2018), there are few studies of European long-term trends of tropospheric $O_3$ from satellite retrievals. Ebojie et al. (2016) found a non-significant negative trend of -0.9 ± 0.5 % yr$^{-1}$ for southern Europe between 2003 – 2011 using tropospheric column data from the Scanning Imaging Absorption Spectrometer for Atmospheric Chartography (SCIAMACHY). Pope et al. (2018) found no significant

trends between 2005 – 2015 across England and Wales for the sub-column $O_3$ (0 – 6 km) from the Ozone Monitoring Instrument (OMI) retrieved by the RAL scheme. However, they found a significant positive $O_3$ trend in Scotland (representing background $O_3$), of 0.172 Dobson units (DU) yr$^{-1}$.

The number of studies of long-term variation in European free tropospheric $O_3$, e.g. from other measurement techniques such

as ozonesondes and aircraft, is fairly limited and provides a mixed picture. From ozonesondes launched from a European site, Oltmans et al. (2013) found $O_3$ in the 500 – 700 hPa layer to have increased from the beginning of the 1970s to the end of the 1980s, and to have then decline slowly to 2010. They found a trend of between ~ 3 – 5 % decade$^{-1}$ at the surface – 300 hPa for 1970 – 2010, but near-zero trends when only 1980 – 2010 is considered. Logan et al. (2012) showed increasing $O_3$ from regular aircraft measurements (from the Measurement of OZone by Airbus In-service airCraft' (MOZAIC) programme) during the

1990s, and showed that the ozonesondes, high-altitude alpine surface sites and aircraft agree on decreasing $O_3$ since 1998.



Gaudel et al. (2018) found little change in ozonesonde observations above southern France 1994 – 2013. The In-service Aircraft for a Global Observing System (IAGOS) commercial aircraft monitoring network highlighted $O_3$ increases in winter (11% increase) and autumn (5% increase) above Frankfurt, Germany (300 – 1000 hPa) in a comparison of 1994 – 1999 and 2009 – 2013, but little change in spring and summer (Gaudel et al., 2018). Two recent studies looking across the whole of

Europe found quite similar results in trends of median $O_3$. Gaudel et al. (2020) found a small trend between 1994 - 2016 from aircraft observations of $1.3 \pm 0.2$ ppbv decade$^{-1}$ (2.4 %) for 700 – 300 hPa; and Christiansen et al. (2022) found trends of between ~ -1 to 4 ppb decade$^{-1}$ across 7 European ozonesonde sites from 1990 – 2017 in the free troposphere, with an average of $1.9 \pm 1.1$ ppb decade$^{-1}$ ($3.4 \pm 2.0\%$ decade$^{-1}$).

Here, we study three RAL UV satellite records (Global Ozone Monitoring Experiment (GOME), SCIAMACHY and OMI) in detail between 1996 – 2017, exploring long-term trends of lower tropospheric column ozone (LTCO$_3$ – surface to 450 hPa) across Europe. We make comparisons using a 3-D chemical transport model (TOMCAT) to provide a common framework for comparing the impact of different sampling and vertical sensitivity between the instruments. We also present trends for the ozonesonde record and TOMCAT simulated tropospheric $O_3$ across the study period. Lastly, we use model experiments to

identify the relative impacts of meteorology and emissions on the model trends across Europe.

## 2 Methods

### 2.1 RAL UV Satellite Data Products

We use three records of satellite LTCO$_3$ from the RAL UV scheme (Miles et al., 2015; Munro et al., 1998). The scheme provided the first satellite retrievals of tropospheric $O_3$ (Munro et al., 1998) and the subsequent tropospheric $O_3$ products have

been used across a variety of studies including Gaudel et al. (2018). The scheme is based on the standard optimal estimation technique by Rodgers (2000) and is described in detail in Miles et al. (2015). All three satellite products have been adjusted for their ensemble mean biases with respect to ozonesondes as a function of month of year and latitude (30° bins – see Russo et al., (2023)). Applying these corrections is intended to mitigate systematic differences between the three instruments' biases while maintaining their temporal variability and evolution. We select from the dataset only sub-columns with an effective

cloud fraction of < 0.2, a solar zenith angle of < 80°, the retrieval convergence flag set = 1.0 and the normalised cost function of < 2.0. We define the European domain as 30°N – 70°N and 30°E – 45°W.

### 2.1.1 GOME

The GOME instrument was aboard the European Space Agency's (ESA's) second European Remote Sensing Satellite (ERS-2) which was launched in April 1995 and ceased operation in 2011 (Burrows et al., 1999; European Space Agency, n.d.). ERS-

2 had a sun-synchronous and near-polar orbit, with an equator crossing time of 10:30 local solar time (LST). The instrument



had 1-D detector arrays providing spectral sampling across four contiguous bands deployed in nadir across-track scanning mode, a swath width of 960 km, a ground-pixel resolution of 40 km (along-track) × 320 km (across-track) and achieved global coverage in ~ 3 days. GOME measured in the UV-near-IR (NIR) wavelength range (240-790 nm) at a spectral resolution of 0.2 – 0.4 nm (Burrows et al., 1999).

### 2.1.2 SCIAMACHY

The SCIAMACHY instrument was aboard ESA's Envisat which was launched in March 2002 and ceased operation in April 2012 (Bovensmann et al., 1999; Ebojie et al., 2016). Envisat had a sun-synchronous and near-polar orbit, with an equator crossing time of 10:00 LST. The instrument had 1-D detector arrays, like GOME, deployed in limb-scanning, nadir-scanning and solar/lunar occultation viewing modes. In nadir-scanning mode it had an across-track width of 960 km and a ground-pixel resolution of 30 km (along-track) × 240 km (across-track). SCIAMACHY measured in the UV- NIR as for GOME and also two SWIR bands spanning a wavelength range (240-2380 nm) at a spectral resolution of 0.2-1.5 nm (Bovensmann et al., 1999).

### 2.1.3 OMI

The OMI instrument is aboard NASA's Aura satellite, launched in July 2004 and is currently still in operation (Levelt et al., 2006). The Aura satellite has a sun-synchronous and near-polar orbit with an equator crossing time of 13:45 LST and flies as part of the 'A-train' formation. The instrument uses a 2-D detector array in a nadir-scanning mode, with the second dimension providing continuous across-track sampling. It has a swath width of 2600 km and a ground resolution of 13 km × 24 km, providing nearly global coverage every day. OMI measures in the UV-Vis wavelength range (270 – 500 nm) with a spectral resolution of 0.45 – 1.0 nm (Levelt et al., 2006). Due to the 2-D detector array of OMI, across-track adjustments were calculated for each detector row and year (relative to an average of all rows), to reduce enhanced stratospheric influence from the longer viewing path at the edges of the swath.

### 2.1.4 Uncertainties

The satellite instruments used here have several known issues e.g. the OMI detector row anomaly (Levelt et al., 2018), the GOME tape recorder failure (Van Roozendael et al., 2012) and UV degradation for GOME-type sensors (eg Miles et al., 2015). Due to the OMI row anomaly, there is a reduction in availability of data from some positions across the swath from ~ 2009 onwards, predominantly from the middle of the swath. To account for this, we have selected rows with consistent seasonal cycle amplitude and shape for the years for which they are available. We calculate across-track adjustments (relative to an average for all rows) for the available rows on a yearly basis to account for the changing number of rows used. The GOME tape recorder failure mostly impacted the southern hemisphere (SH) so had little impact on the European domain used here. To account for UV degradation in GOME and SCIAMACHY, a correction has been applied by RAL prior to the L2 (retrieval) processing step (see Miles et al. (2015)) based on the ratio of UV sun-normalised radiance spectra modelled from a climatology and observed sun-normalised radiances.



As noted above, all three satellite L2 data sets have been compared with ozonesonde ensembles and corrections applied as functions of latitude and month of year to reduce relative systematic errors between the three products (see Russo et al., (2023)).

As in Pope et al. (2015), when multiple soundings are averaged together to form a monthly mean, these random errors will reduce by a factor of $\frac{1}{\sqrt{N}}$ (where N is sample size). We present an estimate of these monthly average random errors across the European domain, scaled according to number of days with filtered retrievals per month to account for averaging. Typically, we find the domain-average monthly random error to be 31.6 (21.4-47.7) %, 31.1 (23.1-44.9) % and 31.5 (21.9-49.3) % for GOME, SCIAMACHY and OMI, respectively. However, when reducing by $\frac{1}{\sqrt{N}}$ (i.e. number of days in the month), this reduces

to 4.1 (2.4-7.2) %, 4.2 (2.8-9.0) % and 2.6 (1.3-4.8) %.

## 2.2 Ozonesondes

We present ozonesonde data from 1996 – 2018 from the World Ozone and Ultraviolet Radiation Data Centre (WOUDC) (WOUDC, 2021), which predominantly uses electrochemical concentration cell (ECC) and Brewer-Mast ozonesondes. The ozonesondes are filtered for records in the European domain and within 3-hours of each satellite overpass time (10:00 and

13:30). The ozonesonde profiles are used to derive LTCO$_3$. Co-located TOMCAT records were produced, using the nearest model grid-box value for each ozonesonde profile.

## 2.3 TOMCAT 3-D Model

TOMCAT is a global 3-D offline chemical transport model forced by ERA-Interim meteorological reanalyses from the European Centre for Medium-Range Weather Forecasts (ECMWF) (Chipperfield, 2006; Dee et al., 2011; Monks et al., 2017).

It has a resolution of 2.8° × 2.8° with 31 vertical levels from the surface and 10 hPa. TOMCAT is coupled with the Global Model of Aerosol Processes (GLOMAP) which calculates aerosol microphysics (Mann et al., 2010; Spracklen et al., 2005). The full chemistry scheme, with 79 species and ~200 chemical reactions, is described in Monks et al. (2017). Anthropogenic surface emissions for NO$_x$, CO and VOCs are from the Coupled Model Intercomparison Project Phase 6 (CMIP6) (Feng et al., 2020). Fixed natural surface emissions (soils/ocean) for NO$_x$, CO and VOCs are from POET (Granier et al., 2005; Olivier et

al., 2003). Fixed annual biogenic emissions of CO and VOCs are from the Chemistry-Climate Model Initiative (CCMI) (Morgenstern et al., 2017). Annual varying biogenic emissions of isoprene and monoterpenes are from the Joint UK Land Environment Simulator (JULES) within the free-running UK Earth System Model (UKESM; Sellar et al., 2019) from a CMIP6 historical setup (Clark et al., 2011; Sellar et al., 2019). Biomass burning emissions are from the Global Fire Emissions Database (GFED) version 4 (van der Werf et al., 2017). Aerosol surface emissions (sulphur dioxide, black carbon, organic carbon) are

from MACCity (Granier et al., 2011). Global average surface TOMCAT methane (CH$_4$) is scaled to the annually varying global average surface CH$_4$ value from National Oceanic and Atmospheric Administration (NOAA) (Dlugokencky, 2020), while retaining its simulated spatial distribution due to emissions and sinks.





Using TOMCAT, we simulated tropospheric $O_3$ between 1996 – 2018, with one year of spin-up. We present satellite records
with two different equator crossing overpass times, approximately 10:00 for Envisat (SCIAMACHY) and ERS-2 (GOME)
and approximately 13:30 for Aura (OMI), and therefore the simulations here have been setup to sample 3-D fields of the model
daily at these two LSTs. This control configuration of TOMCAT is labelled TC-CTL. To identify the relative impact of surface
emissions and meteorology on long-term trends, we performed two model experiments. One experiment used a fixed year of
monthly surface emissions (from 2008, around the mid-point of the study period) and varying meteorology (TC-FX-EMS),
and the other used a fixed year of meteorology reanalyses to force the model (from 2008) and varying surface emissions (TC-
FX-MET). For all TOMCAT simulations, we calculate and present LTCO$_3$. For cases where there is comparison with the
satellite records, the TOMCAT simulations are co-located with the satellite records and have averaging kernels (AKs) applied.

A tracer ($O_{3S}$) for stratosphere-troposphere exchange (STE) is used from the TOMCAT simulations to understand the impact
of $O_3$ transport from the stratosphere. The tracer is set equal to the model-calculated $O_3$ in the stratosphere. When the tracer
enters the troposphere, there are no additional sources of the tracer, as the only tropospheric source of the tracer is transport
from the stratosphere. In the troposphere all sink reactions for $O_3$ apply, e.g. photolysis and reaction with OH and HO$_2$, as well
as dry deposition (Monks et al., 2017). Any $O_3$ that is transported into the stratosphere will be labelled as stratospheric before
it returns. Note that the model does not contain any specific treatment of stratospheric chemistry (e.g. chlorine, bromine, polar
stratospheric clouds) as it uses a climatological ozone vertical boundary condition. However, the flux of air between the lower-
stratosphere to upper-troposphere is well represented in the model (i.e. driven by meteorological reanalyses).

**2.4 Trend Model**

We use the following trend model with a seasonal component, as shown in **Equation 1**:

$$Y_t = C + BX_t + A\sin(\omega X_t + \phi) + N_t \qquad\qquad \textbf{Equation 1}$$

where $Y_t$ is the monthly sub-column $O_3$ for month $t$, C is the sub-column $O_3$ for the first month of the record, $X_t$ is the number
of months after the first month of the record, $A\sin(\omega X_t + \phi)$ is the seasonal component ($A$ is the amplitude, $\omega$ is the
frequency (the period is set to 1 year, $\omega = \frac{\pi}{6}$) and $\phi$ is the phase shift). $N_t$ represents the model errors/residuals unexplained
by the fit function, including interannual variability. C, B, A and $\phi$ represent the fit parameters which are based on a linear
least squares fit. This trend model is based on a function in Weatherhead et al. (1998) and has been used in several studies
looking at long-term trends in tropospheric species (e.g. van der A et al. (2006), van der A. et al., (2008) and Pope et al. (2018,
2024)). Weatherhead et al. (1998) give a derivation for the precision of the trend as a function of the autocorrelation, the length
of the timeseries (in months) and the variance in the fit residuals. The trend precision, $\sigma_B$, is calculated by Equation 2:



$$\sigma_B \approx \left[ \frac{\sigma_N}{n^{\frac{3}{2}}} \sqrt{\frac{(1 + \alpha)}{(1 - \alpha)}} \right]$$

**Equation 2**

where $n$ is the number of years in the record, $\alpha$ is the autocorrelation in the residuals and $\sigma_N$ is the standard deviation in the residuals. In this study, trends are presented in DU yr$^{-1}$ (and % yr$^{-1}$) with ± the precision ($\sigma_B$).

## 3 Results

### 3.1 Long-term Satellite Records

LTCO$_3$ from three satellite records (GOME, SCIAMACHY and OMI) between 1996 – 2018 is shown in Error! Reference s ource not found.. The three records show a LTCO$_3$ seasonal cycle with higher values (around 25.0-30.0 DU) in summer and lower values (around 15.0-20.0 DU) in the winter, and an average seasonal 'amplitude' of 9.6, 10.8 and 11.7 DU for GOME, SCIAMACHY and OMI, respectively. SCIAMACHY and GOME show a large variation between years in seasonal cycle
'amplitude' (difference between maximum and minimum month for each year), with a standard deviation of ~2.2 and 1.8 DU, respectively, whereas OMI shows a smaller variation, with a standard deviation of ~1.1 DU. Broadly, the GOME LTCO$_3$ timeseries indicates an underlying decrease from 1996 – 2002/2003 and then a stabilisation to 2010. For SCIAMACHY, the LTCO$_3$ record is relatively consistent, but shows two large peaks (> 25.0 DU) in the summers of 2007 and 2008. In contrast, the OMI LTCO$_3$ record shows a distinctive pattern over the record, decreasing towards 2009, increasing towards ~ 2015, and
afterwards beginning to decrease again. The three satellite records have six overlapping years (2005 – 2010), which allows for a direct intercomparison (Error! Reference source not found.(**b**)). Across these overlapping years, GOME and SCIAMACHY s how similar LTCO$_3$ absolute values, aside from the higher summer values in 2007 and 2008 for SCIAMACHY, with an average difference of 0.5 DU (**Figure 1**). The OMI values are on average 4.5 DU larger than GOME and 3.9 DU larger than SCIAMACHY. This is despite having applied adjustments to each record based on the mean differences with respective
ozonesonde ensembles (i.e. see Russo et al., (2023) and Pope et al., (2024) for details). Although there is a moderate absolute offset between them, OMI and SCIAMACHY show a high correlation (Pearson's correlation co-efficient (r) = 0.91), with lower correlations between GOME and SCIAMACHY (r = 0.62) and GOME and OMI (r = 0.64) due to GOME's LTCO$_3$ seasonal cycle lagging the other two instruments by approximately 1-month. During the overlap years, GOME has the seasonal cycle with lowest amplitude: 8.5 DU in comparison to 11 DU for SCIAMACHY and OMI. It is notable that UV degradation
of the GOME instrument in this latter period of its operational lifetime had been substantial, giving rise to low optical throughput, correspondingly low signal-to-noise and to a large correction becoming necessary to sun-normalised UV radiances. The differences found here between the instruments could also be due to several other factors, such as overpass time (e.g. diurnal variation in boundary layer thickness and/or O$_3$ mixing ratios), spatial sampling and vertical sensitivity (itself a function of signal-to-noise).




Here, the model acts as a useful framework to investigate the impact of satellite vertical sensitivity on retrieved $LTCO_3$ quantities (e.g. absolute values, trends) but also a further constraint on European $LTCO_3$ spatio-temporal evolution. Thus, the TOMCAT-simulated tropospheric $O_3$ record has been co-located with each satellite retrieval and convolved by the AKs shown in **Equation 3**:

$$mod_{AK} = AK(mod_{int} - apr_{hi}) + apr_{low} \qquad \textbf{Equation 3}$$

where $\textbf{\textit{mod}}_{AK}$ is the vector of modified model sub-columns (Dobson Units, DU), $\textbf{\textit{AK}}$ is the averaging kernel matrix and $\textbf{\textit{mod}}_{int}$ is the vector of model sub-columns (DU) on the satellite retrieval pressure grid. Here, the $\textbf{\textit{AK}}$ is rectangular ($23 \times 19$ levels), so the apriori has two forms: 1) $\textbf{\textit{apr}}_{hi}$ representing 23 sub-columns and 2) $\textbf{\textit{apr}}_{low}$ representing 19 sub-column levels.

Direct comparisons of GOME and SCIAMACHY with the TOMCAT show a model overestimation, of approximately 4.0 DU. and correlation coefficients of 0.72 and 0.82, respectively (**Figure 2**). In contrast, for OMI the model underestimates, with a mean difference of -1.1 DU. Applying the AKs to the model record improves the agreement with GOME and SCIAMACHY broadly decreasing the model $LTCO_3$ record by approximately 2.6 DU in both cases and improving the correlation coefficients from 0.72 to 0.74 for GOME and from 0.83 to 0.92 for SCIAMACHY. For OMI, the application of the AKs also decreases
the sub-column values, however this increases the underestimate by the model, from 1.1 DU to 4.0 DU, and it causes a reduction in the correlation from 0.83 to 0.78.

### 3.2. Satellite, Model and Ozonesonde LTCO₃ Trends

We present European domain-wide trends for the three satellite instruments, the ozonesonde records and co-located model records (with and without AKs applied for the satellite comparisons) (**Table 1**). Across the 15-year GOME record, there is a
negative trend of $-0.21 \pm 0.05$ DU yr$^{-1}$ ($-1.05 \pm 0.26$ % yr$^{-1}$). This negative trend is not captured in the model record, with TOMCAT, both with/without GOME AKs applied, showing a near-zero trend. There is a small negative trend of $-0.20 \pm 0.14$ DU yr$^{-1}$ ($-1.03 \pm 0.26$ % yr$^{-1}$) across the 8-year SCIAMACHY record. This trend is also not captured in the model record, as TOMCAT, both with/without SCIAMACHY AKs applied, shows near-zero trends. For GOME and SCIAMACHY, although neither TOMCAT with/without AKs captures the satellite trends, applying the AKs does subtly change the simulated trend
towards that of the satellite, i.e. makes the model trend more negative. For OMI, there is a near-zero trend across its 13-year record, although there is large inter-annual variability within this period. Interestingly, this near-zero trend is not captured by the model, as TOMCAT with OMI AKs applied shows a negative trend of $-0.26 \pm 0.07$ DU yr$^{-1}$ ($-1.30 \pm 0.37$ % yr$^{-1}$), due to low values around 2014 – 2018. In contrast, the TOMCAT record without AKs applied shows a near-zero trend across this period, as does the OMI $LTCO_3$ record itself. Other studies of free tropospheric $O_3$ trends from observations have found that
these trends are not captured by a model (e.g. Parrish et al. (2014), Young et al. (2018) and Christiansen et al. (2022)). Model trend underestimates (i.e. in terms of magnitude) may be due to uncertainties in prescribed precursor gas emissions and model representation of STE. Christiansen et al. (2022) and Pope et al., (2023a) found dynamics (e.g. STE) to be the more important process controlling the spatio-temporal evolution of free-tropospheric ozone within models, while precursor emissions are



more important at the surface. Overall, the satellite records here suggest a small reduction in lower-tropospheric $O_3$ in the early part of the record, which has then stabilised towards the end of the record.

The European ozonesonde record (for 10:00 ± 3 hours and 13:30 ± 3 hours) between 1996 – 2018 show a near-zero trend for both local time intervals, of 0.01 ± 0.01 DU yr$^{-1}$ and 0.02 ± 0.01 DU yr$^{-1}$, respectively (Error! Reference source not found.). This near-zero trend is captured in the co-located TOMCAT records, with trends of 0.01 ± 0.01 DU yr$^{-1}$ for both time ranges, and they show generally good agreement (r = 0.90 for both time ranges). These near-zero trends are smaller than ozonesonde trends presented in Christiansen et al. (2022) for Europe (~ +3% decade$^{-1}$). However, as well as a different selection of sondes used, their ozonesonde record starts in the early 1990s, a time period which several studies found positive trends in the free troposphere, e.g. Logan et al. (2012), before the stabilisation after ~ 2000. TOMCAT (not co-located) shows a similar trend to the ozonesondes, with a near-zero trend between 1996 – 2018. This near-zero trend is present despite surface emissions of precursor gases used in the model decreasing during this time period (not-shown).

During their overlap period (2005-2010), LTCO$_3$ trends for all three satellite instruments are negative (Error! Reference source not found.**2**), with GOME showing the lowest (-0.17 DU yr$^{-1}$, -0.92 % yr$^{-1}$), SCIAMACHY showing the largest (-0.47 DU yr$^{-1}$, -2.43 % yr$^{-1}$) and OMI showing a value in between (-0.36 DU yr$^{-1}$, -1.55 % yr$^{-1}$). The corresponding ozonesonde trends at the two overpass times (not shown) between 2005 and 2010 are negligible. The model captures the negative satellite trends across the overlap period more convincingly than across their respective complete time periods (Error! Reference source not found.). The model record co-located to GOME shows a very similar negative trend of -0.16 DU yr$^{-1}$ (-0.78 % yr$^{-1}$). For SCIAMACHY and OMI, the co-located model records show smaller negative trends than the satellite, with -0.12 DU yr$^{-1}$ (-0.56 % yr$^{-1}$) and -0.19 DU yr$^{-1}$ (-0.96 % yr$^{-1}$), respectively. These negative trends are ~25% and ~50% the size of the SCIAMACHY and OMI trends, respectively.

### 3.3. Satellite LTCO$_3$ Spatial Trends

As seen in **Tables 1** & **2**, the satellite LTCO$_3$ trends tend to be more consistent over the 2005-2010 period. GOME (1996-2010) has a consistent negative trend of approximately -0.15 DU yr$^{-1}$ across the European domain (**Figure 3**). For SCIAMACHY (2003-2010), the trend is more strongly negative at -0.5 to -0.3 DU yr$^{-1}$ apart from a positive trend (0.2-0.3 DU yr$^{-1}$) in the south of the domain over northern Africa. OMI (2005-2018) shows near-zero trends across large portions of the domain (0.1 DU yr$^{-1}$ 60-70°N and -0.1 DU yr$^{-1}$ 50-60°N) and moderately positive trends (0.2-0.4 DU yr$^{-1}$) over the Mediterranean and northern Africa (predominantly from changes in precursor emissions – see **Figure 5**). For the 2005-2010 period, the LTCO$_3$ trend is consistently negative for SCIAMACHY (<-1.0 to -0.3 DU yr$^{-1}$) and OMI (-0.5 to -0.2 DU yr$^{-1}$) apart from some small positive trends over northern Africa. Therefore, the regional trends **in Table 1** & **2** are broadly representative of most parts of the domain. For GOME, there is again a region of positive trend over northern Africa. Elsewhere





the trend is typically -0.4 to -0.2 DU yr$^{-1}$ although there is more noise in the spatial distribution with a scatter of positive trends (0.1-0.3 DU yr$^{-1}$). Thus, the negative trend for the domain as a whole is smaller (-0.17 DU yr$^{-1}$) than for the other two instruments (-0.47 and -0.36 DU yr$^{-1}$).

## 3.4 Model Experiments

We present two additional TOMCAT simulations, one with a repeating fixed year of emissions (TC-FX-EMS) and the other with a repeating fixed year of meteorology (TC-FX-MET), both using the fixed year of either monthly surface emissions or 6-hour meteorological fields from 2008. We selected 2008 as it represented an approximate mid-point in the study time-period. Both simulations closely represent the control (r = 0.98/0.99 for TC-FX-MET/TC-FX-EMSTC-FX-EMS), with TC-FX-EMS on average 0.41 DU larger and TC-FX-MET 0.16 DU smaller (**Figure 4(a)**). As TC-FX-EMS is larger than the control, this suggests that 2008 was a year of surface emissions which caused higher $O_3$ concentrations than usual, whereas the meteorology of 2008 (used in TC-FX-MET) is more like an average of the whole time period, as shown by the smaller difference with the control. The monthly anomalies for TC-FX-EMS are very similar to the control (**Figure 4(b)**, r = 0.88), highlighting the importance (less importance) of varying meteorology (emissions) in explaining short-term monthly tropospheric $O_3$ variation. The TC-FX-MET simulation is less well correlated with the control (r = 0.57), thus again showing the importance of meteorological variability in controlling tropospheric ozone. However, periods do exist where the emissions dominate in importance such as 1998 (potentially linked to the strong El Nino that year) where the TC-FX-EMS (TC-FX-MET) run struggles (reasonably) captures the control simulation anomaly.

The two fixed simulations show similar near-zero trends to the control between 1996 – 2018 (Error! Reference source not found.). The anomalies of all three simulations show a broadly similar pattern over the time period (**Figure 4(b)**), with moving from negative to more positive anomalies between 1996 and ~2006 – 2008, and then a move from positive to more negative anomalies for the remainder of the time period (~2006 – 2008 to 2018). For this first time period (1996 – 2008) the simulations show very small positive trends, with +0.07 DU yr$^{-1}$ (+0.30 % yr$^{-1}$) for the control, and approximately half the magnitude for both fixed simulations (Error! Reference source not found.). There is a similar story for the second time period (2008 – 2018) but with very small negative trends, with -0.07 DU yr$^{-1}$ (-0.31 % yr$^{-1}$) in the control, and again, approximately half the magnitude for both fixed simulations. This suggests that across both time periods, emissions and meteorology are having a similar influence on the long-term trends, rather than a large cancellation of processes. Therefore, the near-zero trend in the control is not due to a cancellation of trends from a large impact of either emissions or meteorology, despite the reduction in key $O_3$ precursors, e.g. $NO_x$ and VOCs, used in the model.

Spatially, the three simulations show very small trends for each grid-box, ranging from -0.04 to 0.05 DU yr$^{-1}$ (**Figure 5**). TOMCAT (control) shows negative trends across central continental Europe, with the largest negative values around Italy and the Balkans, and also across the northern Atlantic region. Positive trends are found across the southern Atlantic region, the





southern Mediterranean counties in northern Africa and NE Europe. The region of negative trends in the control run over

central continental Europe and positive trends in the southern Atlantic region and southern Mediterranean/North Africa are

present in the TC-FX-MET (varying emissions) only and are therefore regions where the long-term trend is dominated by

changes in surface emissions from the land. The negative trends in the northern Atlantic region, the North Sea and western

Scandinavia and positive trends across the NE of Europe are present in TC-FX-EMS (varying meteorology) only and are

regions where the trend is dominated by changes in meteorology.

STE can also impact tropospheric $O_3$ variation and therefore could influence long-term trends. The simulations use a fixed

climatological value of stratospheric $O_3$ at 10 hPa, but the flux of STE and transport of $O_3$ into the troposphere varies between

the years. Monthly anomalies of a sub-column derived from STE $O_3$ contribution varied between -0.9 and +0.7 DU (or ~ -42%

and +21%) between 1996 – 2018 (**Figure 6**). $O_3$ from STE broadly follows a similar monthly anomaly pattern to the

tropospheric sub-column $O_3$. There is a near-zero trend in the $O_3$ from STE across the time period (0.00 ± 0.01 DU yr$^{-1}$),

indicating that although STE has impacted year-to-year $O_3$ variability, there is no strong trend in the simulated STE flux, that

has influenced tropospheric $O_3$ in the TOMCAT simulations.

## 4 Conclusions

We present a detailed analysis of three satellite products between 1996 - 2017, demonstrating the information they can provide

about long-term trends in lower-tropospheric $O_3$ above Europe. We compare these records with simulated tropospheric $O_3$

from the 3-D chemical transport model TOMCAT and independent measurements of the free troposphere using ozonesondes.

For the GOME (1996 – 2010), SCIAMACHY (2003 – 2010) and OMI (2005 – 2017) lower-tropospheric $O_3$ records there are

negative trends of -0.21 ± 0.05 DU yr$^{-1}$ (-1.05 ± 0.26 % yr$^{-1}$), -0.20 ± 0.14 DU yr$^{-1}$ (-1.03 ± 0.26 % yr$^{-1}$) and a near-zero trend

of 0.00 ± 0.04 DU yr$^{-1}$ (0.00 ± 0.16% yr$^{-1}$), respectively. Overall, there appears to have been a decrease in lower-tropospheric

$O_3$ over Europe from the mid-1990s to early 2000s before a stabilisation in the late 2000s and 2010, and an increase over the

Mediterranean and N. Africa in that latter period (consistent with recent studies e.g. Pope et al., 2023b and 2024). Despite

reasonable agreement with the satellite records, co-located TOMCAT model records do not capture these small trends, showing

predominantly a near-zero trend across the time periods. In contrast to the satellites, observations from the troposphere from

ozonesondes agree with the TOMCAT record in showing a near-zero across the time-period (1996 – 2018).

The three satellite records are compared during their 6-year overlap period (2005 – 2010), showing consistent negative trends

(-0.17 to -0.47 DU yr$^{-1}$), despite there being a systematic off-set in the OMI record (~4 DU larger). During this period, the co-

located model records show greater consistency with those of the satellites, indicating that considering the vertical sensitivity

and spatial sampling does not fully account for the differences seen between the records. The model and ozonesonde trends at



the GOME/SCIAMACHY and OMI mid-morning and early afternoon overpasses also suggest that the different diurnal overpasses between the sensors is not a major contributor to differences in detected $LTCO_3$ trends. Similar results were reported by Pope et al., (2024) but they did not explore the longer-term records of these multiple sensors between 1996 to 2018 (nor use the RAL GOME and SCIAMACHY records). Therefore, additional factors are likely contributing to the instrument differences such as the OMI row anomaly and UV degradation in GOME and SCIAMACHY not being accounted for sufficiently well.

Overall, we have used satellite, ozonesonde and model data to investigate long-term trends in European lower tropospheric ozone. While there is some agreement between the satellite instruments (i.e. moderate negative trends), especially in the overlapping years, the model (with and without the satellite averaging kernels, AKs, applied) and ozonesonde records suggest negligible tendencies. Model sensitivity experiments also suggest that spatiotemporal variability in processes (i.e. precursor emissions, meteorology, and the stratosphere-tropospheric flux) controlling lower tropospheric ozone have remained stable. As a result, it is difficult to detect a robust and consistent linear trend in European lower tropospheric $O_3$ between 1996 and 2017, which is masked by large inter-annual variability in the model and ozonesonde records and especially the UV sensor records. Future trend analyses will benefit for example from new data versions. For example, RAL's scheme has been improved in preparation for full mission re-processings of these and other satellite UV sounders and application to Sentinel-5 Precursor and upcoming Sentinels-4 and -5 which are planned to extend the record to mid-2040s. Additionally, future modelling work including a more complete description of lower stratospheric ozone and chemical budgets would be beneficial.

**Acknowledgements**

This work was funded by the UK Natural Environment Research Council (NERC) by providing funding for the National Centre for Earth Observation (NCEO, award reference NE/R016518/1) and the NERC Panorama Doctoral Training Programme (DTP, award reference 580 NE/S007458/1). The TOMCAT runs were undertaken on ARC3, part of the High-Performance Computing facilities at the University of Leeds, UK.

**Data Availability**

The GOME, SCIAMACHY and OMI data are available via the NERC Centre for Environmental Data Analysis (CEDA) Jasmin platform subject to data requests. However, these satellite datasets and the TOMCAT model simulations will be uploaded to an open access repository like Zenodo should this manuscript be published in ACP.

**Author Contributions**

MAP and RJP conceptualised, planned and undertook the research study. BJK, RS and BGL provided the data and advice on using the products. MAP performed the TOMCAT model simulations with support from MPC and WF. MAP and RJP prepared the manuscript with contributions from all co-authors.

**Conflicts of Interest**

The authors declare no conflicts of interest.



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

**Figures and Tables:**

| Observational/Model Records | Trend (DU yr$^{-1}$; %) | Trend 95% CI (DU yr$^{-1}$) | Trend p-value |
|---|---|---|---|
| GOME (1996-2010) | -0.21 (-1.05) | (-0.31, -0.11) | 0.00 |
| TC-GOME (1996-2010) | 0.06 (0.26) | (0.02, 0.10) | 0.00 |
| TC-GOME+AK (1996-2010) | -0.01 (-0.04) | (-0.05, 0.03) | 0.80 |
| SCIAMACHY (2003-2010) | -0.20 (-1.03) | (-0.47, 0.07) | 0.14 |
| TC-SCIAMACHY (2003-2010) | 0.00 (0.00) | (-0.11, 0.11) | 1.00 |
| TC-SCIAMACHY+AK (2003-2010) | -0.01 (-0.05) | (-0.15, 0.13) | 1.00 |
| OMI (2005-2017) | 0.00 (0.00) | (-0.08, 0.08) | 1.00 |
| TC-OMI (2005-2017) | -0.26 (-1.30) | (-0.41, 0.11) | 0.00 |
| TC-OMI+AK (2005-2017) | -0.07 (-0.30) | (-0.13, -0.01) | 0.02 |
| Ozonesondes T1 (1998-2018) | 0.01 (0.05) | (-0.01, 0.03) | 0.32 |
| TC-Ozonesondes T1 (1998-2018) | 0.01 (0.04) | (-0.03, 0.05) | 0.62 |
| Ozonesondes T2 (1998-2018) | 0.02 (0.09) | (0.00, 0.04) | 0.32 |
| TC-Ozonesondes T2 (1998-2018) | 0.01 (0.03) | (-0.03, 0.05) | 0.62 |
| TC-CTL (1996-2018) | -0.01 (-0.03) | (-0.04, 0.02) | 0.50 |

**Table 1**: Satellite, ozonesonde and model LTCO$_3$ trends (DU yr$^{-1}$ and % yr$^{-1}$) for their respective time periods. For each satellite instrument
(GOME, SCIAMACHY and OMI) the co-located model records with and without AKs applied are also presented. The ozonesonde trends are presented for two local time intervals (T1 = 10.00 LT and T2 = 13.30 LT) and with co-located model records. CI = confidence interval. Here, TC = TOMCAT and CTL = the TOMCAT control simulation. The trend p-values are also shown.







| Observational/Model Records | Trend (DU yr$^{-1}$; %) | Trend 95% CI (DU yr$^{-1}$) | Trend p-value |
|---|---|---|---|
| GOME (2005-2010) | -0.17 (-0.92) | (-0.22, -0.12) | 0.00 |
| TC-GOME (2005-2010) | -0.01 (-0.02) | (-0.03, 0.01) | 0.32 |
| TC-GOME+AK (2005-2010) | -0.16 (-0.78) | (-0.14, -0.12) | 0.00 |
| SCIAMACHY (2005-2010) | -0.47 (-2.43) | (-0.61, -0.33) | 0.00 |
| TC-SCIAMACHY (2005-2010) | -0.05 (-0.21) | (-0.10, 0.00) | 0.32 |
| TC-SCIAMACHY+AK (2005-2010) | -0.12 (-0.56) | (-0.19, -0.05) | 0.00 |
| OMI (2005-2010) | -0.36 (-1.55) | (-0.40, -0.32) | 0.00 |
| TC-OMI (2005-2010) | -0.07 (-0.31) | (-0.10, -0.04) | 0.00 |
| TC-OMI+AK (2005-2010) | -0.19 (-0.96) | (-0.26, -0.12) | 0.00 |

**Table 2**: Satellite, ozonesonde and model LTCO$_3$ trends for 2005-2010. For each satellite instrument (GOME, SCIAMACHY and OMI) the co-located model records with the AKs applied are also presented. The ozonesonde trends are presented for two local time intervals (T1 = 10.00 LT and T2 = 13.30 LT) and with co-located model records. CI = confidence interval. The trend p-values are also shown.

| Observational/Model Records | Trend (DU yr$^{-1}$; %) | Trend 95% CI (DU yr$^{-1}$) | Trend p-value |
|---|---|---|---|
| TC-CTL (1996-2018) | -0.01 (-0.03) | (-0.04, 0.02) | 0.50 |
| TC-EMS (1996-2018) | 0.00 (0.00) | (-0.02, 0.02) | 1.00 |
| TC-MET (1996-2018) | 0.00 (0.00) | (-0.02, 0.02) | 1.00 |
| TC-CTL (1996-2008) | 0.07 (0.30) | (0.06, 0.08) | 0.00 |
| TC-EMS (1996-2008) | 0.03 (0.12) | (0.02, 0.04) | 0.00 |
| TC-MET (1996-2008) | 0.04 (0.17) | (0.03, 0.05) | 0.00 |
| TC-CTL (2008-2018) | -0.07 (-0.31) | (-0.08, -0.06) | 0.00 |
| TC-EMS (1996-2008) | -0.03 (-0.13) | (-0.04, -0.02) | 0.00 |
| TC-MET (1996-2008) | -0.04 (-0.17) | (-0.05, -0.03) | 0.00 |
| TC-STE (1996-2018) | 0.00 (0.00) | (-0.01, 0.01) | 1.00 |

**Table 3**: Model LTCO$_3$ trends (DU yr$^{-1}$ and % yr$^{-1}$) for 1996-2018, 1996-2008 and 2008-2018 from the TC-CTL, TC-EMS and TC-MET simulations. CI = confidence interval. TC-STE is the TOMCAT tracer for the stratospheric ozone flux into the tropopsphere calculated as the LTCO$_3$. The trend p-values are also shown.




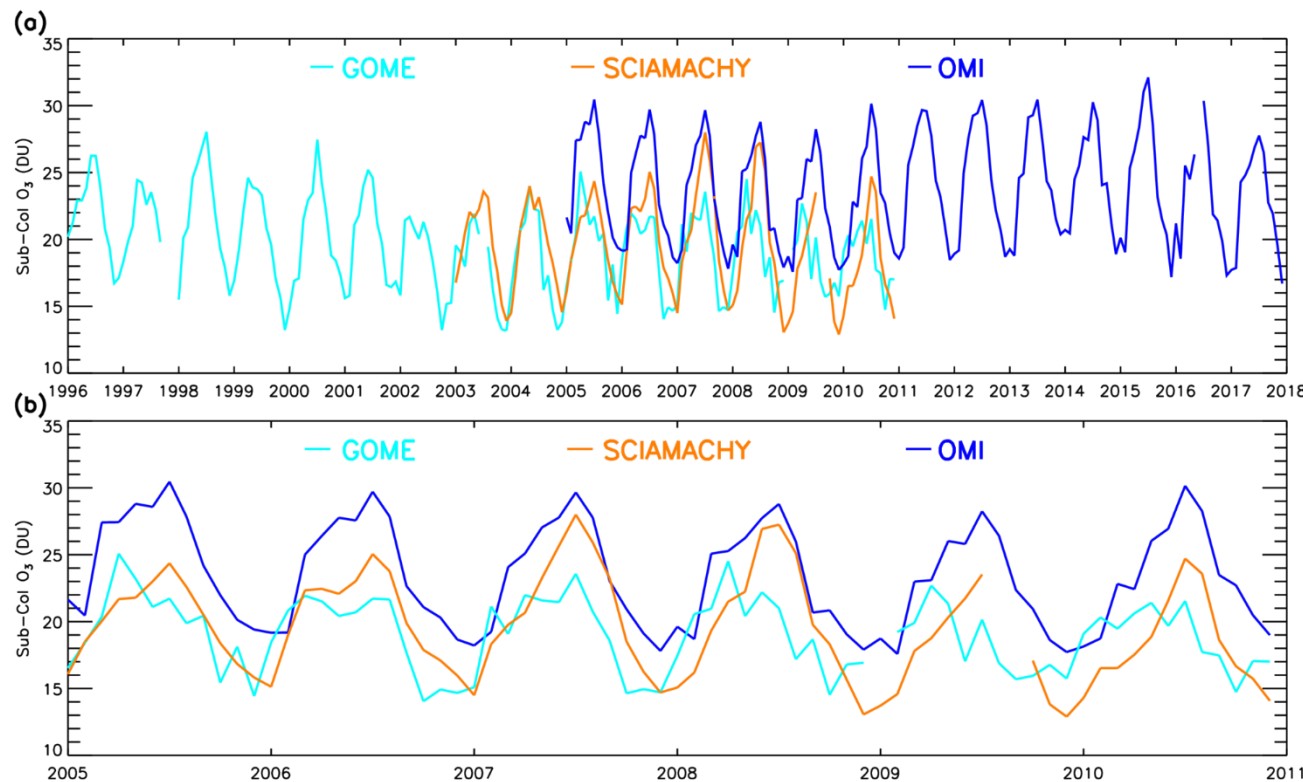


**Figure 1:** Timeseries of European monthly average satellite lower tropospheric column $O_3$ (LTCO$_3$, surface-450 hPa) (DU) between (a) 1996 – 2017 full record and (b) 2005 – 2010 overlap period. The Pearson's correlation coefficient (r), 2005-2010, between GOME-SCIAMACHY, GOME-OMI and SCIAMACHY-OMI time-series are 0.62, 0.64, 0.91, respectively. The average differences are -0.5, -4.5 and -3.9 DU, respectively.






**Figure 2:** Timeseries of European monthly average LTCO$_3$ (DU) between 1996 – 2017 for (a) GOME, (b) SCIAMACHY and (c) OMI. Co-located model records (with and without AKs applied) for each satellite record are also shown. The correlations and mean differences between the model and respective satellite records (TOMCAT-satellite, DU) are shown at the top of each panel.






**Figure 3:** LTCO$_3$ trends (DU yr$^{-1}$) for each grid-box of the European sub-column satellite records for (a) GOME (1996 – 2010), (b) GOME (2005 – 2010), (c) SCIAMACHY (2003 – 2010), (d) SCIAMACHY (2005-2010), (e) OMI (2005-2018) and (f) OMI (2005 – 2010).



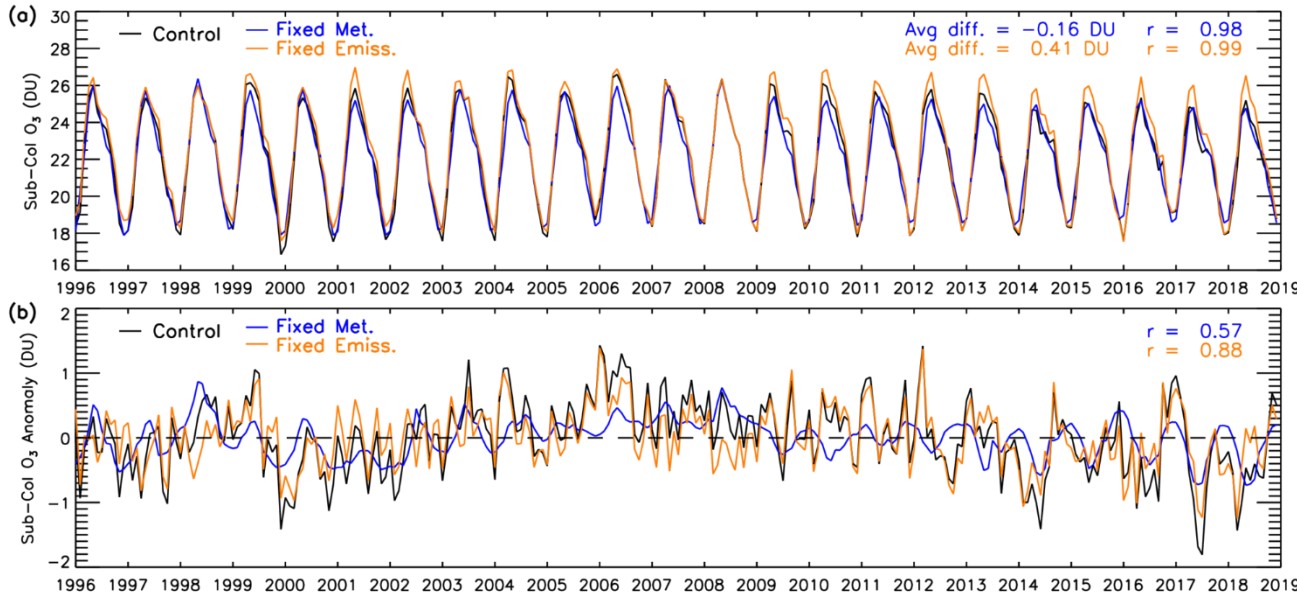

**Figure 4**: (a) Timeseries of European average monthly LTCO$_3$ trends for TOMCAT (control), TC-FX-EMS and TC-FX-MET between 1996 – 2018. The average difference and *r* of the two experiments and the control is presented in the top right of the panel. (b) Monthly mean anomalies (relative to a 1996 – 2018 baseline) for the 3 simulations (DU).



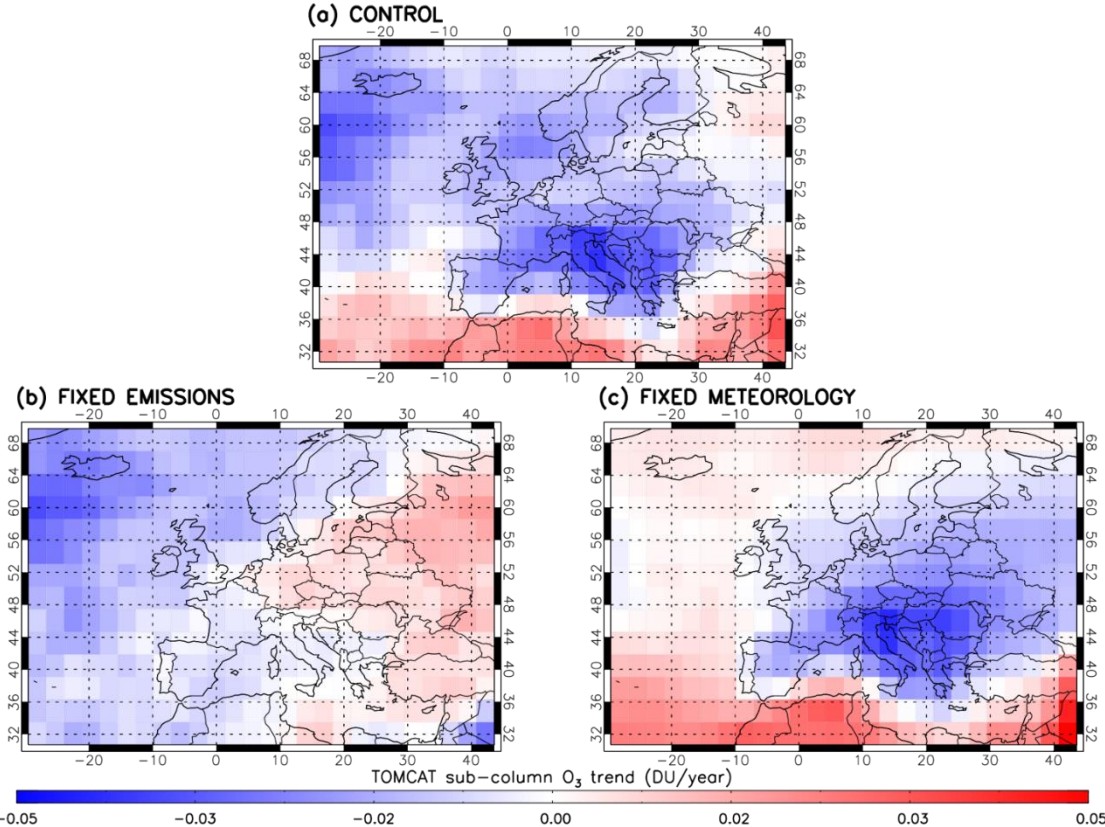

**Figure 5:** LTCO$_3$ trends across the European domain (DU yr$^{-1}$) for (a) TC-control, (b) TC-FX-EMS and (c) TC-FX-MET.




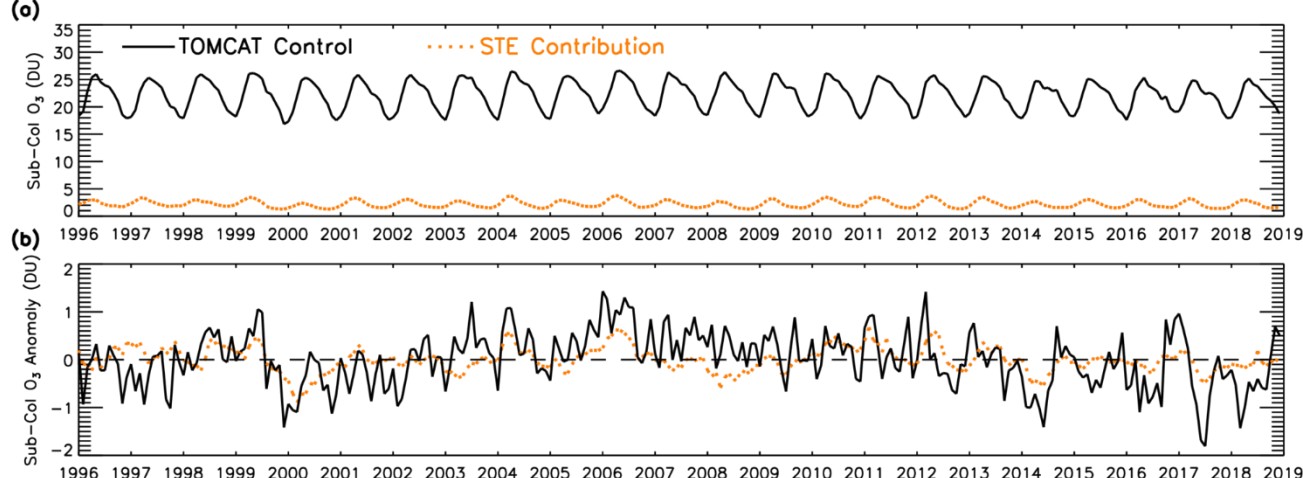

**Figure 6:** (a) Timeseries of European monthly average LTCO$_3$ and STE LTCO$_3$ (DU) time-series from 1996 – 2018 (DU). (b) Absolute anomalies for both records (DU) (relative to a baseline of 1996 – 2018).
