# Peer review of "Long-term satellite trends of European lower-tropospheric ozone from 1996-2017"

_EGUsphere, 2024_

## Community Comment (CC1)

February 2, 2025

Comments by Owen R. Cooper (TOAR Scientific Coordinator of the Community Special Issue) on:

**Long-term satellite trends of European lower-tropospheric ozone from 1996 – 2017**

Matilda A. Pimlott, Richard J. Pope, Brian J. Kerridge, Richard Siddans, Barry G. Latter, Wuhu Feng, Martyn P. Chipperfield

EGUsphere [preprint], https://doi.org/10.5194/egusphere-2024-3717
Discussion started:  16 Dec 2024
Discussion closes:   5 Feb 2025

This review is by Owen Cooper, TOAR Scientific Coordinator of the TOAR-II Community Special Issue. I, or a member of the TOAR-II Steering Committee, will post comments on all papers submitted to the TOAR-II Community Special Issue, which is an inter-journal special issue accommodating submissions to six Copernicus journals:  ACP (lead journal), AMT, GMD, ESSD, ASCMO and BG. The primary purpose of these reviews is to identify any discrepancies across the TOAR-II submissions, and to allow the author teams time to address the discrepancies.  Additional comments may be included with the reviews. While O. Cooper and members of the TOAR Steering Committee may post open comments on papers submitted to the TOAR-II Community Special Issue, they are not involved with the decision to accept or reject a paper for publication, which is entirely handled by the journal's editorial team.

**Comments regarding TOAR-II guidelines:**

TOAR-II has produced two guidance documents to help authors develop their manuscripts so that results can be consistently compared across the wide range of studies that will be written for the TOAR-II Community Special Issue.  Both guidance documents can be found on the TOAR-II webpage:
https://igacproject.org/activities/TOAR/TOAR-II

*The TOAR-II Community Special Issue Guidelines*:   In the spirit of collaboration and to allow TOAR-II findings to be directly comparable across publications, the TOAR-II Steering Committee has issued this set of guidelines regarding style, units, plotting scales, regional and tropospheric column comparisons, and tropopause definitions.

*The TOAR-II Recommendations for Statistical Analyses*:  The aim of this guidance note is to provide recommendations on best statistical practices and to ensure consistent communication of statistical analysis and associated uncertainty across TOAR publications. The scope includes approaches for reporting trends, a discussion of strengths and weaknesses of commonly used techniques, and calibrated language for the communication of uncertainty. Table 3 of the TOAR-II statistical guidelines provides calibrated language for describing trends and uncertainty, similar to the approach of IPCC, which allows trends to be discussed without having to use the problematic expression, "statistically significant".

**Detailed comments:**

Lines 59-73

      The review of past ozone trend studies for Europe needs to be updated to include the most recent work, including a new TOAR-II study. While the papers by Oltmans et al. (2013) and Logan et al. (2012) are very good, they are now quite out of date, and aren't really relevant for current ozone trends.

      An important consideration for trend detection is sample size. Since 1988 there have been quite a few studies that have examined the impact of ozonesonde sampling frequency on our ability to detect ozone trends in the free troposphere, as reviewed by the recent TOAR-II paper by Chang et al. (2024). Chang et al. (2024) also provide extensive analysis demonstrating that sparse temporal sampling by ozonesondes, especially once per week sampling, often fails to detect an ozone trend.

      Estimates of free tropospheric ozone trends above Europe vary widely, based on the available ozonesonde time series. For example, free tropospheric trends (700-300 hPa) for the period 1994-2019 range from -1.56 ± 0.85 ppbv per decade (p-value<0.01) above Payerne, Switzerland, to +2.26 ± 1.04 ppbv per decade (p-value<0.01) above De Bilt, The Netherlands (Chang et al., 2022). To improve our ability to detect trends, Chang et al. (2022) developed a method to merge all available IAGOS and ozonesonde time series above western Europe, and calculated a regional trend of 0.60 ± 0.20 ppbv per decade (p-value<0.01) for the tropospheric column (950-250 hPa) and a regional trend of 0.65 ± 0.19 ppbv per decade (p-value<0.01) for the free troposphere (700-300 hPa).

      A new paper submitted to the TOAR-II Community Special Issue (Van Malderen et al., 2024) builds on the work of Chang et al. (2022), and shows clear ozone increases above Europe based on a merged dataset of ozonesonde, IAGOS, FTIR, Umkehr and lidar time series (stations are listed in Table 1, and additional time series information is provided in Table S1 in the supplement; Table 2 lists the data coverage, and Figure 3 shows the domain). Figure 12 and Table S2 (supplement) show that the 1995-2019 tropospheric column (surface to 8 km) regional ozone trend for Europe is 0.47 ± 0.20 ppbv per decade (*p*-value<0.01); the authors assign a confidence level of very high to this trend. Trends in the free troposphere (700-300 hPa) are even stronger: 1.25 ± 0.27 ppbv per decade (p-value<0.01); the authors assign a confidence level of very high to this trend.

      The findings from Chang et al. (2022) and Van Malderen et al. (2024), along with the earlier findings of Gaudel et al. (2020), Christiansen et al. (2022) and Wang et al. (2022), all demonstrate that ozone has increased above Europe since the mid-1990s. It would be helpful if you can summarize this well documented ozone increase in the Introduction, and use it to frame the findings from your study.

Line 136

Here the manuscript introduces the ozonesondes that are used to evaluate the satellite products and the model, but I couldn't find any information on the actual time series that were chosen. To understand the ozonesonde analysis, and to judge if it supports the conclusions, the reader needs the following basic information: number of stations used, names of stations, coordinates of stations, sampling rate, instruments flown, time period analyzed, and finally, were the time series treated individually or merged? Van Malderen et al. (2024) is a very good reference for presenting this type of information. The authors mention that the ozonesondes are selected to match satellite overpass times of 10:00 and 13:30 LST. Presumably if sondes fall outside of the time window then they are discarded. What does this do to the sample sizes? Ozonesonde sampling rates are already too low for accurate trend detection, and if data are thrown out then the ability to accurately detect a trend diminishes even further.

In addition to using ozonesondes, why not also use the IAGOS profiles from Frankfurt? This is the only location in the world where ozone is profiled multiple times per day, and as a result, it is the only location in the world where we have high confidence that the monthly mean profile is accurate (we also

have high confidence in the monthly distribution, i.e. 5th to 95th percentiles).  As a result, the ozone trend at this location is also highly accurate:  +1.16 ± 0.77 ppbv per decade (p-value<0.01) in the free troposphere for the period 1994-2019 (Chang et al., 2022).

Many of the authors on this paper are also authors on a paper recently published in the TOAR-II Community Special Issue that merges the GOME, SCIAMACHY and OMI products into a single tropospheric ozone product (Pope et al., 2023).  Yet, as far as I can tell, the current paper makes no mention of the merged product, and treats the GOME, SCIAMACHY and OMI products separately. Why not include the merged product?

Line 359-361
The paper concludes with this statement:
"As a result, it is difficult to detect a robust and consistent linear trend in European lower tropospheric O3 between 1996 and 2017, which is masked by large inter-annual variability in the model and ozonesonde records and especially the UV sensor records."
Regarding the in situ ozone trends, as described above, studies that use merged data sets with very large sample sizes are able to detect a positive trend above Europe for the period 1995-2019.  The trend is there, but if the sample size is too small, the trend cannot be detected.  As no details regarding your ozonesonde analysis were provided, I cannot judge why you were not able to detect an ozone trend, but my guess is that you are using time series with low sampling rates.

Figure 1
Your Figure 1 shows the month to month changes in ozone for the three satellite products from 1996 to 2017. Figure 9 from Chang et al. (2022) (pasted below) shows the monthly ozone values above Europe for the same time period, but based on merged IAGOS and ozonesonde data.  It would be helpful to describe the variability of your satellite products in relation to these in situ observations.

[Figure]

From Chang et al. (2022):  "Figure 9. Quantified annual ozone mean anomalies (with 2-sigma intervals) and uncertainty weighted time series in the free troposphere (700-300 hPa) above Europe and western North America."

**References**

Chang, K.-L., O. R. Cooper, A. Gaudel, M. Allaart, G. Ancellet, H. Clark, S. Godin-Beekmann, T. Leblanc, R. Van Malderen, P. Nédélec, I. Petropavlovskikh, W. Steinbrecht, R. Stübi, D. W. Tarasick, C. Torres (2022), Impact of the COVID-19 economic downturn on tropospheric ozone trends: an uncertainty weighted data synthesis for quantifying regional anomalies above western North America and Europe, AGU Advances, 3, e2021AV000542. https://doi.org/10.1029/2021AV000542

Chang, K.-L., Cooper, O. R., Gaudel, A., Petropavlovskikh, I., Effertz, P., Morris, G., and McDonald, B. C. (2024), Technical note: Challenges in detecting free tropospheric ozone trends in a sparsely sampled environment, Atmos. Chem. Phys., 24, 6197–6218, https://doi.org/10.5194/acp-24-6197-2024

Christiansen, A., Mickley, L. J., Liu, J., Oman, L. D. and Hu, L.: Multidecadal increases in global tropospheric ozone derived from ozonesonde and surface site observations: Can models reproduce ozone trends?, Atmos. Chem. Phys., 22(22), 14751–14782, doi:10.5194/acp-22-14751-2022, 2022.

Gaudel, A., et al. (2018), Tropospheric Ozone Assessment Report: Present-day distribution and trends of tropospheric ozone relevant to climate and global atmospheric chemistry model evaluation, Elem. Sci. Anth., 6(1):39, DOI: https://doi.org/10.1525/elementa.291

Gaudel, A., O. R. Cooper, K.-L. Chang, I. Bourgeois, J. R. Ziemke, S. A. Strode, L. D. Oman, P. Sellitto, P. Nédélec, R. Blot, V. Thouret, C. Granier (2020), Aircraft observations since the 1990s reveal increases of tropospheric ozone at multiple locations across the Northern Hemisphere. Sci. Adv. 6, eaba8272, DOI: 10.1126/sciadv.aba8272

Gulev, S.K., P.W. Thorne, J. Ahn, F.J. Dentener, C.M. Domingues, S. Gerland, D. Gong, D.S. Kaufman, H.C. Nnamchi, J. Quaas, J.A. Rivera, S. Sathyendranath, S.L. Smith, B. Trewin, K. von Schuckmann, and R.S. Vose, 2021: Changing State of the Climate System. In Climate Change 2021: The Physical Science Basis. Contribution of Working Group I to the Sixth Assessment Report of the Intergovernmental Panel on Climate Change [Masson-Delmotte, V., P. Zhai, A. Pirani, S.L. Connors, C. Péan, S. Berger, N. Caud, Y. Chen, L. Goldfarb, M.I. Gomis, M. Huang, K. Leitzell, E. Lonnoy, J.B.R. Matthews, T.K. Maycock, T. Waterfield, O. Yelekçi, R. Yu, and B. Zhou (eds.)]. Cambridge University Press, Cambridge, United Kingdom and New York, NY, USA, pp. 287–422, doi:10.1017/9781009157896.004

Pope, R. J., Kerridge, B. J., Siddans, R., Latter, B. G., Chipperfield, M. P., Feng, W., Pimlott, M. A., Dhomse, S. S., Retscher, C., and Rigby, R.: Investigation of spatial and temporal variability in lower tropospheric ozone from RAL Space UV–Vis satellite products, Atmospheric Chemistry and Physics, 23, 14 933–14 947, https://doi.org/10.5194/acp-23-14933-2023, 2023

Van Malderen, R., Z. Zang, K.-L. Chang, R. Bjorklund, O. R. Cooper, J. Liu, C. Vigouroux, E. Maillard Barras, I. Petropavlovskikh, T. Leblanc, V. Thouret, P. Wolff, P. Effertz, A. Gaudel, H.G.J. Smit, A. M. Thompson, R. M. Stauffer, D. E. Kollonige, D. Tarasick, et al. (2024), Ground-based Tropospheric Ozone Measurements: Regional tropospheric ozone column trends from the TOAR-II/ HEGIFTOM homogenized datasets, submitted to ACP (TOAR-II Community Special Issue), https://doi.org/10.5194/egusphere-2024-3745

Wang, H., Lu, X., Jacob, D. J., Cooper, O. R., Chang, K.-L., Li, K., Gao, M., Liu, Y., Sheng, B., Wu, K., Wu, T., Zhang, J., Sauvage, B., Nédélec, P., Blot, R., and Fan, S. (2022), Global tropospheric ozone trends, attributions, and radiative impacts in 1995–2017: an integrated analysis using aircraft (IAGOS) observations, ozonesonde, and multi-decadal chemical model simulations, *Atmos. Chem. Phys., 22*, 13753–13782, https://doi.org/10.5194/acp-22-13753-2022

---

## Author Comment (AC2)

**Author Responses to Reviewer Comments**

We thank the reviewers for their useful and constructive comments/feedback. We also thank Owen Cooper for his useful comments on our manuscript in relation to the TOAR-II special edition. We have reproduced their comments below in black text, followed by our responses in red text. Please note, where appropriate, we have number listed the reviewer comments for clarification. Any additions to the manuscript are in blue text and our reference to line numbers is based on the originally submitted manuscript.

**Reviewer #1's Comments:**

*Top Level Comments:*

Pimlott et al. present an analysis of the trend of low altitude ozone (surface – 450 hPa) over Europe for the period 1996-2017. They compare three satellite datasets, GOME, SCIMACHY and OMI, and use a chemical transport model, TOMCAT, to assess the role of changing emissions and variable meteorology in driving the trend.

The paper is well written and presents a useful comparison of different satellite ozone products. Its content is suitable for ACP. The relatively limited scope and clear description of the study means I have few comments. Overall, I recommend it for publication after the following issues have been dealt with.

We thank Reviewer #1 for their constructive overview of our manuscript.

*General Comments:*

1.  Line 220: While averaging kernels are an important part of satellite-model comparison and known to the remote sensing field, there are many in the modelling field who know little of them. More detail should be given as to how they work and why they can alter the trend so substantially (e.g. GOME).

Reviewer #1 makes a good point. We have added some additional information on Page 3 Line 92 to provide more information and motivation on the satellite AKs.

"The averaging kernels (AKs), provided with RAL Space' ozone products from satellite UV-Vis nadir sounders, provide the vertical sensitivities of the different layers retrieved with  of the optimal estimation approach applied to the respective instruments (as discussed in Miles et al., (2015)). Pope et al., (2023) provides a detailed assessment of the RAL Space AKs (e.g. Figures 1 and 2 of that study) finding that peak tropospheric $O_3$ sensitivity is in the lower tropospheric layer (surface – 450 hPa), which is the focus of this study. To allow direct like-for-like comparisons of models (e.g. TOMCAT) with these satellite data sets, AKs (i.e. for each layer, essentially a vertical weighting of the retrieval sensitivity) need to be mapped onto the modelled vertical profile before comparable quantities (e.g. LTCO$_3$)  can be compared. Here, we use the TOMCAT model as a tool to help investigate the impact of the AKs (i.e. vertical sensitivity) on satellite derived LTCO$_3$ trends over Europe (i.e. how substantially do the satellite AKs influence the simulated LTCO$_3$ trends).".

2.  The change in emissions over the period of interest should be described in more detail and shown graphically to give the reader an idea of the (relative) magnitude of the changes in key species including NOx and CO. Spatially variation should also be considered as different parts of Europe are likely to have different temporal variations in emissions.

We have included a new Figure showing the TOMCAT NO$_x$ and CO trends between 1996 and 2017.

[Figure]

**Figure 2**: Average TOMCAT emissions (Gg) between 1996 and 2017 for a) $NO_x$ and b) carbon monoxide (CO). TOMCAT emission trends (Gg/year) between 1996 and 2017 for c) $NO_x$ and d) CO. Green polygon-outlined regions show substantial emission trends with p-values < 0.05.

On Page 6 Line 163 we have added the following discussion:

"**Figure 2** shows the surface emissions for $NO_x$ and carbon monoxide (CO), key $O_3$ precursor gases, and their tendencies between 1996 and 2017. The $NO_x$ (**Figure 2a**) and CO (**Figure 2b**) emission long-term averages peak over northwestern Europe at >10 Gg and >100 Gg per grid box. The corresponding trends (**Figure 2c** and **2d**) show substantial (p-value < 0.05 – green polygon-outlined regions) decreases of < -5.0 Gg/year and <-10 Gg/year. Therefore,…..".

3. While I do not wish to create substantially more work for the authors, an interesting additional experiment might be to fixed say the emissions over the Po Valley and surrounding regions, where the strongest modelled reduction in O3 is simulated, and allow emissions in other regions to vary to determine the relative influence of local O3 production and longer range transport.

Reviewer #1 makes an interesting point, and in principle we agree that additional model simulations for Po valley and other regions e.g. northern  Africa and Turkey would be an interesting research aim. In practice, however, such additional model simulation would be outside the scope of this paper. We have updated the following sentence in the Conclusions on Page 12 Line 361-362 from:

"Future trend analyses will benefit for example from new data versions." to

"Future trend analyses will benefit from planned new satellite data versions and potentially also some additional model sensitivity experiments (fixing regional emissions for e.g. the Po Valley which

has the largest impact in the TC-FX-MET experiment, though relatively modest in absolute LTCO$_3$ trend terms).".

4. I notice that there are several missing references which are denoted with the text: "*Error! Reference source not found.*"

Well spotted. We found the issues on (and changed to):

Page 7 Line 192 – "Figure 1"

Page 7 Line 201 – "Figure 1"

Page 9 Line 253 – "Table 1"

Page 9 Line 261 – "Table 2"

Page 9 Line 266 – "Table 2"

Page 10 Line 300 – "Table 2"

Page 10 Line 305 – "Figure 4a"

5. The complexity of the O3 satellite data for those outside the remote sensing community means I would recommend that the data used for this study are uploaded to the referenced Zenodo repository along with the model data.

We fully support this suggestion. As stated on Page 12 Lines 370-371, we say "*However, these satellite datasets and the TOMCAT model simulations will be uploaded to an open access repository like Zenodo should this manuscript be published in ACP*". Therefore, if this manuscript is accepted for publication in ACP, we will do this as suggested by the reviewer.

**Reviewer #2's Comments:**

*Top Level Comments:*

Pimlott et al. investigated long-term trends (1996-2017) in tropospheric ozone over Europe using measurements from satellite instruments (GOME, SCIAMACHY, OMI) and ozonesondes, as well as simulations from a global chemical transport model (TOMCAT). The trend analysis is thorough and the information is presented very clearly. My comments mainly relate to the significance of the results and their interpretation within a broader context.

We thank Reviewer #2 for their constructive overview of our manuscript.

*General comments:*

1. Given that the major outcome of this work is a near-zero trend in tropospheric ozone across the period of interest, why is this outcome significant? Can we draw any major conclusions about the processes controlling ozone during this period? Any insights into a future outlook for tropospheric ozone? I understand that the authors have explored some of the underlying processes using sensitivity simulations with TOMCAT but the overall significance of the results found in this work could be addressed more directly.

In reply to the reviewer's question: "why is this outcome significant" we would contend that near-zero trends for tropospheric ozone (LTCO$_3$) over Europe for the study period, is as significant a finding as would have been a positive or negative trend. While we have shown that LTCO$_3$ in the Europe-centred region examined remained relatively stable in this time period, our model sensitivity experiments have shown that simulated trends for both fixed emissions and fixed meteorology are

negligible, so it's *not* that the impact of emissions and meteorology have cancelled each other out to form a near-zero trend. Therefore, the trends in LTCO$_3$ are stable because the major processes controlling it have near-zero trends and year-year variability dominates.

In the conclusions, we have now made this clearer on Page 12 Lines 354-361:

"*Overall, we have used satellite, ozonesonde and model data to investigate long-term trends in European lower tropospheric ozone. While there is some agreement between the satellite instruments (i.e. modest negative trends), especially in the overlapping years, the model (with and without the satellite averaging kernels, AKs, applied) and ozonesonde records suggest negligible tendencies. Model sensitivity experiments also suggest that spatiotemporal variability in processes (i.e. precursor emissions, meteorology, and the stratosphere-tropospheric flux) controlling lower tropospheric ozone have remained stable. As a result, it is difficult to detect a robust and consistent linear trend in European lower tropospheric O$_3$ between 1996 and 2017, which is masked by large inter-annual variability in the model and ozonesonde records and especially the UV-Vis sensor records.*".

2. More background information is needed in the introduction to put this work into a broader context and to assist in the interpretation of results. For example, the authors did mention that anthropogenic ozone precursor emissions declined over the period of interest, but this is only one piece of the puzzle - ozone chemistry and its dependence on precursor emissions is not linear (which in itself may offer a possible explanation for a near-zero trend in recent years) and this must be communicated in the manuscript. Precursor emissions are similarly complex and can be anthropogenic but have other sources too, including vegetation and biomass burning, and this should be discussed as well. I would also like to see some discussion of how meteorology affects tropospheric ozone as this is a major process investigated in the sensitivity simulations conducted by the authors towards the end of the manuscript.

The aim of this study was not to investigate individual processes which would influence tropospheric O$_3$ trends (e.g. biomass burning emissions or temperature). More generally, it aimed to identify trends in the broader factors controlling O$_3$. Therefore, we focussed on fixing the emissions and meteorology for the year 2008 to investigate the impacts. In terms of the emissions, in line with Reviewer #1's general comment #2, we have provided some more analysis on the total emissions. Anthropogenic sources make up the bulk of the emissions (e.g. as shown in Table 1 of Monks et al., (2017)), so vegetations fires and wildfires are going to have a secondary or third order impact on the emission results, which showed near-zero trends anyway. In one of our previous studies (Pope et al., 2023), we undertook a detailed assessment of how emissions, meteorology, deposition and stratospheric ozone fluxes influenced European tropospheric ozone. While not presented in the manuscript, we did some sensitivity experiments to isolate the impact of fire emissions on tropospheric O$_3$ and the impacts were very small.

Overall, the study by Pope et al., (2023) was able to investigate the impacts of individual processes as we were only focussing on 2017 and 2018. However, to repeat all of these sensitivity experiments (approximately 6 or 7 runs in total) for 20+ years would not be practical and would likely provide limited information beyond what the current sensitivity experiments have shown. Therefore, while we appreciate the additional information the reviewer is trying to get us to investigate/discuss, it is realistically outside the scope of this study.

However, we have added the following text on Page 6 Line 166:

"While there are many non-linear processes controlling the spatiotemporal evolution of tropospheric $O_3$ (e.g. temperature, advection, deposition, photochemistry, stratosphere-troposphere exchanges, different precursor emission sources (e.g. anthropogenic, wildfires)), it is not practical in this study to undertake a 22-year sensitivity experiment for each process. Pope et al., (2023) did undertake a detailed assessment of factors contributing to the European summer 2018 tropospheric $O_3$ event, focussing on 2017 and 2018. Therefore, we limited ourselves to the fixed emissions and meteorology experiments between 1996 and 2017. Here, the fixed meteorology experiment refers to influences of meteorological variables like temperature, cloud cover (i.e. influence on photochemistry) and then long-range transport (e.g. advection of $O_3$-rich air masses).".

***Minor comments:***

1. Line 128: Please provide more information here about the ozonesonde corrections: How are these corrections applied? How much do they impact the total reported values? Do the corrections differ between the three satellite products? Understood that references are provided but this seems worth addressing directly in the manuscript, especially if the corrections are sizeable or if there's a chance that the corrections could contribute to differences between the three satellite products during the period of overlap.

Several other studies also go into the process of comparing the satellite data with the ozonesondes, derivation of the bias corrections and their impact in greater detail. Therefore, instead of reproducing all that information again here, we feel it is more beneficial to add the additional references, which answer these questions in detail. These are Pope et al., (2023b) and Pope et al., (2024a).

2. Line 133: How are these error values calculated? Any idea how the RAL products compare to other retrievals that may exist for the same instruments?

The satellite errors are determined in the optimal estimation approach which derives from that used by Miles et a., (2015) and applied to generate all the RAL Space datasets. We use the random and systematic errors directly from the product files. The RAL data sets have been directly compared with ozonesondes, along with other products, in Keppens et al., (2018). They found that RAL Space LTCO$_3$ products from various UV-Vis sounders had positive biases of several 10s% compared with the ozonesondes in the LTCO$_3$; lowest at 10% for OMI. On the other hand, an IASI product, from a different research group, showed negative biases of 20-30%, so comparable magnitude to that of RAL Space UV products. However, as outlined in the manuscript, application of a correction mitigates a large proportion of the bias with respect to sondes.

To make these points clearer, we have added the following text:

Page 3 Line 86 we have updated "*technique by Rodgers (2000) and is described in detail in Miles et al. (2015).*" to "technique by Rodgers (2000) and is described in detail in Miles et al. (2015), including treatment of errors. Comparison of the RAL Space UV-Vis satellite products with ozonesondes on a wider scale (i.e. Keppens et a., 2018) found a 10-40% positive bias, comparable to the magnitude of other satellite lower tropospheric ozone products in the same study. There, all three…..".

3. Section 2.2: Where are the ozonesondes launched from? An additional figure might be useful here to place their locations within the European domain.

[Figure]

**Figure 1:** European distribution of ozonesondes used in this study and time series of annual ozonesonde frequencies (i.e. all sites and times).

We have added an additional figure (new Figure 1) of the ozonesonde launch sites. Please see the new figure above. We have added the following text on Page 5 Line 141:

"The location of the European ozonesondes (and annual frequency) used in this study is shown in **Figure 1**.".

4. Line 225 and elsewhere: Why include TOMCAT results where the averaging kernels are not applied? My understanding is that AKs are applied to align the vertical sensitivity between the model and the satellite measurements, and that the two datasets are not truly comparable until this is done.

The satellite AKs are applied to TOMCAT to allow direct like-with-like comparisons between the model and the observations. It also allows us to quantify the impact the satellite vertical sensitivity on the observed trends by assessing the impact in has on the simulated trends. However, we report the model tends with and without the AKs applied, so when we want to undertake analysis of the processes controlling the simulated trends (i.e. sensitivity experiments), we need to analyse the original model runs. To make this clearer we have added the following text Page 6 Line 167:

"Where we are inter-comparing model simulations (or with the ozonesondes, which have no issues with vertical sensitivity as they are in-situ measurements with high vertical resolution), application of the satellite AKs to the model is not required.".

5. Section 3.3: Could a near-zero overall trend in tropospheric ozone be due to a cancelling of larger positive and negative trends in different locations across the domain? Precursor

emissions may be declining in Europe but what trends are coming from Africa? What could be driving those trends? Overall, transport from Africa seems to be an important contributing factor and could be developed further in the interpretation of results.

As displayed in Figure 4, there will be some cancellation of regional trends in the domain average LTCO$_3$ trends. However, the largest absolute trends in individual pixels peak at 0.05 DU/yr, which is comparable to the domain average trends reported in Table 3. So, it looks like cancellation of regional trends is generally a secondary issue.

To make this clear in the manuscript, we have added the following on Page 11 Line 320:

"Overall, there is likely some cancellation of regional tendencies in the domain average LTCO$_3$ trends for each model experiment (**Figure 4**). However, given the absolute magnitude of these pixel-by-pixel based trends, which are comparable to the overall regional trends (**Table 3**), it will have a limited impact on the big picture as LTCO$_3$ appears to be relatively stable in most (if not all) spatial regions.".

As to the European and African (i.e. regional trends), please see our response to Reviewer #1's general comment #3.

6. Section 3.4: What is meant by "meteorology" here? Which variables specifically did you investigate?

Please see our response to Reviewer #2's general comment #2.

7. Line 358: The authors state "Model sensitivity experiments also suggest that spatiotemporal variability in processes (i.e. precursor emissions, meteorology, and the stratosphere-tropospheric flux) controlling lower tropospheric ozone have remained stable" but this is not consistent with the previous assertion that precursor emissions have decreased over that time.

We would argue that this is correct. From our modelling results, all the sensitivity experiments, including the fixed-emissions run, suggest they have negligible impact on the LTCO$_3$ trends. The emissions will likely have more of an impact at the surface, but in the lower free troposphere, the impact is less important. A similar thing was found by Pope et al., (2023) where meteorological processes appeared to be dominating the O$_3$ variability. However, the long-term meteorological tendency seems to have limited impact here. Therefore, we believe this statement is correct and our results support this assertion.

8. Table 3: Some of the entries in the first column appear to be mislabeled (TC-EMS (1996-2008) and TC-MET (1996-2008) appear twice).

Yes, this is incorrect. The second occurrences of TC-MET and TC-EMS should be for 2008-2018. This has now been corrected.

9. Throughout: Seems to be some inconsistency in using 2017 vs 2018 to represent the final year of the long-term record.

The satellite records covered the period of 1996 to 2017 (if all combined). However, we have the model and ozonesonde data for an additional year. Since the satellite data are not consistent for the entire record and we primarily focus on 2005-2010 for the satellites, we believe using the additional year of data for the model and ozonesondes is appropriate.

10. Throughout: Errors in some figure reference labels

Please see our response to Reviewer #1' general comment #4.

**Owen Cooper's Comments:**

*Top Level Comments*

1. Comments regarding TOAR-II guidelines: TOAR-II has produced two guidance documents to help authors develop their manuscripts so that results can be consistently compared across the wide range of studies that will be written for the TOARII Community Special Issue. Both guidance documents can be found on the TOAR-II webpage: https://igacproject.org/activities/TOAR/TOAR-II

Thank you for making us aware of these resources. We have tried to adhere to these in this manuscript and others we have submitted to TOAR-II wherever possible.

2. The TOAR-II Community Special Issue Guidelines: In the spirit of collaboration and to allow TOAR-II findings to be directly comparable across publications, the TOAR-II Steering Committee has issued this set of guidelines regarding style, units, plotting scales, regional and tropospheric column comparisons, and tropopause definitions.

Thank you for making us aware of these resources. We have tried to adhere to these in this manuscript and others we have submitted to TOAR-II wherever possible.

3. The TOAR-II Recommendations for Statistical Analyses: The aim of this guidance note is to provide recommendations on best statistical practices and to ensure consistent communication of statistical analysis and associated uncertainty across TOAR publications. The scope includes approaches for reporting trends, a discussion of strengths and weaknesses of commonly used techniques, and calibrated language for the communication of uncertainty. Table 3 of the TOAR-II statistical guidelines provides calibrated language for describing trends and uncertainty, similar to the approach of IPCC, which allows trends to be discussed without having to use the problematic expression, "statistically significant".

Thank you for making us aware of these resources. We have tried to adhere to these in this manuscript and others we have submitted to TOAR-II wherever possible.

*General Comments*

1. Lines 59-73 The review of past ozone trend studies for Europe needs to be updated to include the most recent work, including a new TOAR-II study. While the papers by Oltmans et al. (2013) and Logan et al. (2012) are very good, they are now quite out of date, and aren't really relevant for current ozone trends.

We have updated the Introduction with some of the new studies suggest by Owen Cooper. Please see below the modified paragraph from Page 2 Line 59 to Page 3 Line 73 in the manuscript from:

"The number of studies of long-term variation in European free tropospheric $O_3$ e.g. from other measurement techniques such as ozonesondes and aircraft is fairly limited and provides a mixed story. From ozonesondes launched from a European site, Oltmans et al. (2013) found $O_3$ in the 500 – 700 hPa layer to have increased from the beginning of the 1970s to the end of the 1980s, and to have then declined slowly to 2010. They found a trend of between ~ 3 – 5 % decade$^{-1}$ at the surface – 300 hPa for 1970 – 2010, but near-zero trends when only 1980 – 2010 is considered. Logan et al.

(2012) showed increasing O$_3$ from regular aircraft measurements (from the Measurement of OZone by Airbus In-service airCraft' (MOZAIC) program) during the 1990s, and showed that the ozonesondes, surface high-altitude alpine sites and aircraft agree on decreasing O$_3$ since 1998. Gaudel et al. (2018) found little change in ozonesonde observations above southern France in 1994 – 2013. The In-service Aircraft for a Global Observing System (IAGOS) commercial aircraft monitoring network highlighted O$_3$ increases in winter (11% increase) and autumn (5% increase) above Frankfurt, Germany (300 – 1000 hPa) in a comparison of 1994 – 1999 and 2009 – 2013, but little change in spring and summer  Two recent studies looking across the whole of Europe found quite similar results in trends of median O$_3$. Gaudel et al. (2020) found a small trend between 1994 - 2016 from aircraft observations of 1.3 ± 0.2 ppbv decade$^{-1}$ (2.4 %) for 700 – 300 hPa; and Christiansen et al. (2022) found trends of between ~ -1 to 4 ppb decade$^{-1}$ across 7 European ozonesonde sites from 1990 – 2017 in the free troposphere, with an average of 1.9 ± 1.1 ppb decade$^{-1}$ (3.4 ± 2.0% decade$^{-1}$).” to

“The number of studies of long-term variation in European free tropospheric O$_3$, e.g. from other measurement techniques such as ozonesondes and aircraft, is fairly limited and provides a mixed story. From ozonesondes launched from a European site, Oltmans et al. (2013) found O$_3$ in the 500 – 700 hPa layer to have increased from the beginning of the 1970s to the end of the 1980s, and to have then decline slowly to 2010. They found a trend of between ~ 3 – 5 % decade$^{-1}$ at the surface – 300 hPa for 1970 – 2010, but near-zero trends when only 1980 – 2010 is considered. Logan et al. (2012) showed increasing O$_3$ from regular aircraft measurements (from the Measurement of OZone by Airbus In-service airCraft' (MOZAIC) program) during the 1990s, and showed that the ozonesondes, surface high-altitude alpine sites and aircraft agree on decreasing O$_3$ since 1998. Gaudel et al. (2018) found little change in ozonesonde observations above southern France in 1994 – 2013. The In-service Aircraft for a Global Observing System (IAGOS) commercial aircraft monitoring network highlighted O$_3$ increases in winter (11% increase) and autumn (5% increase) above Frankfurt, Germany (300 – 1000 hPa) in a comparison of 1994 – 1999 and 2009 – 2013, but little change in spring and summer (Gaudel et al., 2018). Two recent studies looking across the whole of Europe found quite similar results in trends of median O$_3$. Gaudel et al. (2020) found a small trend between 1994 - 2016 from aircraft observations of 1.3 ± 0.2 ppbv decade$^{-1}$ (2.4%) for 700 – 300 hPa; and Christiansen et al. (2022) found trends of between ~ -1 to 4 ppb decade$^{-1}$ across 7 European ozonesonde sites from 1990 – 2017 in the free troposphere, with an average of 1.9 ± 1.1 ppb decade$^{-1}$ (3.4 ± 2.0% decade$^{-1}$). Change et al., (2022), using a merged IAGOS-ozonesonde dataset, found positive trends in the free troposphere (700-300 hPa) of 0.63±0.24 ppbv/decade, but in the boundary level (950-800 hPa) there were negligible trends over Europe. Wang et al., (2022) found similar results with weak positive tropospheric ozone trends (<1.0 ppbv/decade) over Europe between 1995 and 2017.”.

2. An important consideration for trend detection is sample size. Since 1988 there have been quite a few studies that have examined the impact of ozonesonde sampling frequency on our ability to detect ozone trends in the free troposphere, as reviewed by the recent TOAR-II paper by Chang et al. (2024). Chang et al. (2024) also provide extensive analysis demonstrating that sparse temporal sampling by ozonesondes, especially once per week sampling, often fails to detect an ozone trend.

Geographical sampling by ozonesondes and their decreasing coverage in recent years are well-known limitations. They are however the most appropriate surface-based record of tropospheric

ozone profiles for use in analyses of decadal scale variability with which to validate and complement the homogeneous global records from satellites over the last three decades. Their coverage is denser over Europe than many regions so many studies have used ozonesonde records over Europe to investigate long-term trends in tropospheric ozone (e.g. Christiansen et al. (2022), Pope et al., (2024), Oltmans et al. (2013), Gaudel et al. ,2018), amongst many more). We therefore believe it appropriate to use ozonesondes in this study.

3. Estimates of free tropospheric ozone trends above Europe vary widely, based on the available ozonesonde time series. For example, free tropospheric trends (700-300 hPa) for the period 1994-2019 range from -1.56 ± 0.85 ppbv per decade (p-value<0.01) above Payerne, Switzerland, to +2.26 ± 1.04 ppbv per decade (p-value < 0.01) above De Bilt. The Netherlands (Chang et al., 2022). To improve our ability to detect trends, Chang et al. (2022) developed a method to merge all available IAGOS and ozonesonde time series above western Europe, and calculated a regional trend of 0.60 ± 0.20 ppbv per decade (p-value < 0.01) for the tropospheric column (950-250 hPa) and a regional trend of 0.65 ± 0.19 ppbv per decade (p-value < 0.01) for the free troposphere (700-300 hPa).

Please see our response to Owen Cooper's general comment #1.

4. A new paper submitted to the TOAR-II Community Special Issue (Van Malderen et al., 2024) builds on the work of Chang et al. (2022), and shows clear ozone increases above Europe based on a merged dataset of ozonesonde, IAGOS, FTIR, Umkehr and lidar time series (stations are listed in Table 1, and additional time series information is provided in Table S1 in the supplement; Table 2 lists the data coverage, and Figure 3 shows the domain). Figure 12 and Table S2 (supplement) show that the 1995- 2019 tropospheric column (surface to 8 km) regional ozone trend for Europe is 0.47 ± 0.20 ppbv per decade (p-value < 0.01); the authors assign a confidence level of very high to this trend. Trends in the free troposphere (700-300 hPa) are even stronger: 1.25 ± 0.27 ppbv per decade (p-value < 0.01); the authors assign a confidence level of very high to this trend.

Please see our response to Owen Cooper's general comment #1. However, the Van Malderen manuscript has not been published yet, so cannot be referenced.

5. The findings from Chang et al. (2022) and Van Malderen et al. (2024), along with the earlier findings of Gaudel et al. (2020), Christiansen et al. (2022) and Wang et al. (2022), all demonstrate that ozone has increased above Europe since the mid-1990s. It would be helpful if you can summarize this well documented ozone increase in the Introduction, and use it to frame the findings from your study.

Please see our response to Owen Cooper's general comment #1.

6. Line 136 Here the manuscript introduces the ozonesondes that are used to evaluate the satellite products and the model, but I couldn't find any information on the actual time series that were chosen. To understand the ozonesonde analysis, and to judge if it supports the conclusions, the reader needs the following basic information: number of stations used, names of stations, coordinates of stations, sampling rate, instruments flown, time period analyzed, and finally, were the time series treated individually or merged? Van Malderen et al. (2024) is a very good reference for presenting this type of information. The authors mention that the ozonesondes are selected to match satellite overpass times of 10:00 and 13:30 LST. Presumably if sondes fall outside of the time window then they are discarded.

What does this do to the sample sizes? Ozonesonde sampling rates are already too low for accurate trend detection, and if data are thrown out then the ability to accurately detect a trend diminishes even further.

Please see our responses to Review #2's minor comment #3 and Owen Cooper's general comment #2.

7.   In addition to using ozonesondes, why not also use the IAGOS profiles from Frankfurt? This is the only location in the world where ozone is profiled multiple times per day, and as a result, it is the only location in the world where we have high confidence that the monthly mean profile is accurate (we also have high confidence in the monthly distribution, i.e. 5th to 95th percentiles). As a result, the ozone trend at this location is also highly accurate: +1.16 ± 0.77 ppbv per decade (p-value<0.01) in the free troposphere for the period 1994-2019 (Chang et al., 2022).

This is an interesting point, but it is only one site. So, it is difficult to know how representative it would be for all of Europe. Granted, the number of ozonesonde sites is limited spatially, and the IAGOS data at this site would improve the temporal coverage. However, the issue of spatial representation from one airport is an unknown. And while we could investigate the full IAGOS data set, the ozonesondes where a secondary resource to investigate the key model-satellite results. Therefore, given the issues just mentioned, we politely suggest that it is beyond the scope of this study to start investigating new data sets at this late stage.

8.   Many of the authors on this paper are also authors on a paper recently published in the TOAR-II Community Special Issue that merges the GOME, SCIAMACHY and OMI products into a single tropospheric ozone product (Pope et al., 2023). Yet, as far as I can tell, the current paper makes no mention of the merged product, and treats the GOME, SCIAMACHY and OMI products separately. Why not include the merged product?

The merged product from Pope et al., (2023) is a level-3 product, so does not allow for the assessment of the individual sets of AKs from each instrument used in this study. In line with our response to Owen Cooper's general comment #1, we have updated some of the literature in the Introduction section.

9.   Line 359-361 The paper concludes with this statement: "As a result, it is difficult to detect a robust and consistent linear trend in European lower tropospheric O3 between 1996 and 2017, which is masked by large inter-annual variability in the model and ozonesonde records and especially the UV sensor records." Regarding the in situ ozone trends, as described above, studies that use merged data sets with very large sample sizes are able to detect a positive trend above Europe for the period 1995-2019. The trend is there, but if the sample size is too small, the trend cannot be detected. As no details regarding your ozonesonde analysis were provided, I cannot judge why you were not able to detect an ozone trend, but my guess is that you are using time series with low sampling rates.

We are focussing on $LTCO_3$ (surface to 450 hPa) which covers a different altitude range to the majority of the studies suggested here by Owen Cooper, thus could be explaining some of the differences. For instance, in Chang et al., (2022), Figure 5, there are positive trends in the free troposphere, but between approximately 950 to 850 hPa, there are weak negative trends, which are also covered by the $LTCO_3$ altitude range. Secondly, while it has been suggested that the ozonesondes may not have sufficient sample sizes to detect trends (please see our response to

Owen Cooper's general comment #2), the model (TOMCAT) covers the full temporal and spatial range and suggests there are negligible lower tropospheric $O_3$ trends (when and when not co-located to the ozonesonde sites). This is supported by the analysis of Pope et al., (2024) using satellite data, ozonesondes and the UKESM model. In Pope et al., (2023), Figure 8 shows the merged $LTCO_3$ product for GOME-SCIAMACHY-OMI, which shows negligible trends and is supported by ozonesonde sites which have data over the period of interest, again showing negligible trends. So, regardless of whether or not the ozonesonde sampling is an issue, satellite data and models have shown consistent negligible trends in $LTCO_3$.

10. Figure 1 Your Figure 1 shows the month to month changes in ozone for the three satellite products from 1996 to 2017. Figure 9 from Chang et al. (2022) (pasted below) shows the monthly ozone values above Europe for the same time period, but based on merged IAGOS and ozonesonde data. It would be helpful to describe the variability of your satellite products in relation to these in situ observations.

[Figure]

From Chang et al. (2022): "Figure 9. Quantified annual ozone mean anomalies (with 2-sigma intervals) and uncertainty weighted time series in the free troposphere (700-300 hPa) above Europe and western North America."

So, Figure 1 of our manuscript represents satellite records for the surface – 450 hPa range. Thus, to be able to directly compare the time-series (Figure 9) from Chang et al., (2022), we would need satellite data columns between 700 and 300 hPa and we would also need to apply the satellite AKs to the observations to allow like-for-like comparisons. Therefore, given the different altitude ranges (which we would argue are quite substantial i.e. includes boundary layer information and less influence from stratospheric intrusion) and the issue of satellite vertical sensitivity, we are not sure how much we can learn from comparing these two figures (i.e. our Figure 1 and Chang et al., (2022)'s Figure 9).

**References:**

Chang, K.-L., O. R. Cooper, A. Gaudel, M. Allaart, G. Ancellet, H. Clark, S. Godin-Beekmann, T. Leblanc, R. Van Malderen, P. Nédélec, I. Petropavlovskikh, W. Steinbrecht, R. Stübi, D. W. Tarasick, C. Torres (2022), Impact of the COVID-19 economic downturn on tropospheric ozone trends: an

uncertainty weighted data synthesis for quantifying regional anomalies above western North America and Europe, AGU Advances, 3, e2021AV000542. https://doi.org/10.1029/2021AV000542.

Monks, S. A., Arnold, S. R., Hollaway, M. J., Pope, R. J., Wilson, C., Feng, W., Emmerson, K. M., Kerridge, B. J., Latter, B. L., Miles, G. M., Siddans, R., and Chipperfield, M. P.: The TOMCAT global chemical transport model v1.6: description of chemical mechanism and model evaluation, Geosci. Model Dev., 10, 3025–3057, https://doi.org/10.5194/gmd-10-3025-2017, 2017.

Pope, R. J., Kerridge, B. J., Chipperfield, M. P., Siddans, R., Latter, B. G., Ventress, L. J., Pimlott, M. A., Feng, W., Comyn-Platt, E., Hayman, G. D., Arnold, S. R., and Graham, A. M.: Investigation of the summer 2018 European ozone air pollution episodes using novel satellite data and modelling, Atmos. Chem. Phys., 23, 13235–13253, https://doi.org/10.5194/acp-23-13235-2023, 2023a.

Pope, R. J., Kerridge, B. J., Siddans, R., Latter, B. G., Chipperfield, M. P., Feng, W., Pimlott, M. A., Dhomse, S. S., Retscher, C., and Rigby, R.: Investigation of spatial and temporal variability in lower tropospheric ozone from RAL Space UV–Vis satellite products, Atmos. Chem. Phys., 23, 14933–14947, https://doi.org/10.5194/acp-23-14933-2023, 2023b.

Pope, R. J., Rap, A., Pimlott, M. A., Barret, B., Le Flochmoen, E., Kerridge, B. J., Siddans, R., Latter, B. G., Ventress, L. J., Boynard, A., Retscher, C., Feng, W., Rigby, R., Dhomse, S. S., Wespes, C., and Chipperfield, M. P.: Quantifying the tropospheric ozone radiative effect and its temporal evolution in the satellite era, Atmos. Chem. Phys., 24, 3613–3626, https://doi.org/10.5194/acp-24-3613-2024, 2024a.

Pope, R. J., O'Connor, F. M., Dalvi, M., Kerridge, B. J., Siddans, R., Latter, B. G., Barret, B., Le Flochmoen, E., Boynard, A., Chipperfield, M. P., Feng, W., Pimlott, M. A., Dhomse, S. S., Retscher, C., Wespes, C., and Rigby, R.: Investigation of the impact of satellite vertical sensitivity on long-term retrieved lower-tropospheric ozone trends, Atmos. Chem. Phys., 24, 9177–9195, https://doi.org/10.5194/acp-24-9177-2024, 2024b.

Wang, H., Lu, X., Jacob, D. J., Cooper, O. R., Chang, K.-L., Li, K., Gao, M., Liu, Y., Sheng, B., Wu, K., Wu, T., Zhang, J., Sauvage, B., Nédélec, P., Blot, R., and Fan, S. (2022), Global tropospheric ozone trends, attributions, and radiative impacts in 1995–2017: an integrated analysis using aircraft (IAGOS) observations, ozonesonde, and multi-decadal chemical model simulations, Atmos. Chem. Phys., 22, 13753–13782, https://doi.org/10.5194/acp-22-13753-2022.

---

## Author Response (AR2)

**Author Responses to Editor Comments**

We thank the Editor for their useful and constructive comments/feedback on the Abstract. We are pleased to hear our manuscript has been accepted to ACP subject to these changes to the Abstract length. Below is a copy of the new abstract (243 words < 250-word ACP limit) in italics.

**Abstract:**

Tropospheric ozone ( $O_3$ ) is a harmful secondary atmospheric pollutant and an important greenhouse gas. Multiple satellite records have shown conflicting long-term O₃ trends across regions of the globe, including Europe. Here, we investigate lower-tropospheric sub-column O3 (LTCO3, surface – 450 hPa) records from three ultraviolet (UV) sounders produced by the Rutherford Appleton Laboratory (RAL): the Global Ozone Monitoring Experiment (GOME, 1996-2010), Scanning Imaging Absorption Spectrometer for Atmospheric Chartography (SCIAMACHY, 2003-2011) and Ozone Monitoring Instrument (OMI, 2005-2017). GOME and SCIAMACHY detect negative trends of approximately -0.2 DU  $yr^{-1}$ , while OMI indicates a negligible trend. The TOMCAT 3-D chemical transport model was used to investigate processes driving simulated trends and identify possible reasons for satellite trend discrepancies The simulated LTCO₃ trends were negligible (consistent with ozonesonde trends), even when spatiotemporally co-located to the satellite level-2 swath data and convolved by averaging kernels (i.e. a measure of the satellite retrieval vertical sensitivity). Model sensitivity experiments with the emissions or meteorology fixed to 2008 also showed negligible LTCO3 trends between 1996 and 2018, indicating that changes in emissions and meteorology had limited impact on LTCO3 temporal evolution. Given the substantial decrease in air pollutant emissions, this was unexpected, while year-to-year variability dominated the meteorological influence on LTCO3. Finally, we find a negligible trend in the long-term stratosphere O₃ flux into the free troposphere over this period arriving over Europe. Overall, our observational and modelling analysis indicates that European LTCO3 trends have been stable between 1996 and 2018.